# The role of tone duration in dichotic temporal order judgment II: Extending the boundaries of duration and age

Leah Fostick[1]*, Harvey Babkoff[2]

**1** Department of Communication Disorders, Ariel University, Ariel, Israel, **2** Department of Psychology, Bar-Ilan University, Ramat-Gan, Israel

* leah.fostick@ariel.ac.il

## Abstract

Temporal order judgment (TOJ) measures the ability to correctly perceive the order of consecutive stimuli presented rapidly. Our previous research suggested that the major predictor of auditory dichotic TOJ threshold, a paradigm that requires the identification of the order of two tones, each of which is presented to a different ear, is the time separating the onset of the first tone from the onset of the second tone (stimulus-onset-asynchrony, SOA). Data supporting this finding, however, was based on a young adult population and a tone duration range of 10–40 msec. The current study aimed to evaluate the generalizability of the earlier finding by manipulating the experimental model in two different ways: a) extending the tone duration range to include shorter stimulus durations (3–8 msec; Experiment 1) and b) repeating the identical testing procedure on a different population with temporal processing deficits, i.e., older adults (Experiment 2). We hypothesized that the SOA would predict the TOJ threshold regardless of tone duration and participant age. Experiment 1 included 226 young adults divided into eight groups (each group receiving a different tone duration) with duration ranging from 3–40 msec. Experiment 2 included 98 participants aged 60–75 years, divided into five groups by tone duration (10–40 msec). The results of both experiments confirmed the hypothesis, that the SOA required for performing dichotic TOJ was constant regardless of stimulus duration, for both age groups: about 66.5 msec for the young adults and 33 msec longer (100 msec) for the older adults. This finding suggests that dichotic TOJ threshold is controlled by a general mechanism that changes quantitatively with age. Clinically, this has significance because quantitative changes can be more easily remedied than qualitative changes. Theoretically, our findings show that, with dichotic TOJ, tone duration affects threshold by providing more time between the onsets of the consecutive stimuli to the two ears. The findings also imply that a temporal processing deficit, at least among older adults, does not elicit the use of a different mechanism in order to judge temporal order.

**Data Availability Statement:** Data was uploaded to Kaggle. https://www.kaggle.com/leahfostick/tone-duration-and-isi-in-toj.

**Funding:** The author(s) received no specific funding for this work.

**Competing interests:** The authors have declared that no competing interests exist.

## Introduction

Temporal order judgment (TOJ) measures the individual's ability to correctly perceive the order of consecutive stimuli presented rapidly. Most TOJ tasks involve the presentation of only two stimuli in order to measure basic perceptual abilities without confounding the task by adding a memory component. The two stimuli are usually separated by a silent interval between the *offset* of the first tone and the *onset* of the second tone, referred to as the inter-stimulus-interval (ISI), which is manipulated throughout the TOJ task. Short ISIs result in a very rapid presentation of the two stimuli, while longer ISIs result in a slower presentation. When the order of two sounds is judged, the two stimuli must differ by at least one dimension to enable identification. As a result, auditory TOJ paradigms use stimuli that differ either in: a) frequency (pitch) [1–14]; or b) spectrum (pure tone vs. noise) [14]; or c) duration [14, 15]; or d) the ear of presentation, i.e., the ear that receives the first and the ear that receives the second stimulus (referred to as dichotic, spatial, or binaural TOJ) [1–3, 5–12, 14, 16–25].

TOJ has been studied extensively, beginning with the seminal work of Hirsh [26] and Hirsh and Sherrick [27] who measured the amount of time between the onsets of two stimuli (tones, clicks, lights, and their combinations) necessary to correctly report their order. This measure, called the TOJ threshold, reflects the minimum amount of time separating the *onsets* of the two stimuli at which an individual can correctly identify the order of stimulus presentation 75% of the time. Hirsh and Sherrick [27] originally reported the threshold for TOJ to be 17 msec, regardless of the type of stimulus and presentation modality used [27]. However, more recent studies have, generally, reported longer thresholds [1–3, 5–7, 10–12, 14, 16–17, 20, 23– 25].

In a previous study [1], we and others reported the sensitivity of the dichotic TOJ paradigm to methodological and stimulus parameters, specifically to stimulus duration. We considered the possibility that the two manipulations, tone duration and ISI, might affect perception differently, since increasing tone duration increases the amount of sound—thus, the amount of *stimulation*—at the two ears, while increasing the ISI increases the silent interval between these stimulations—i.e., the *lack of stimulation*. However, we found that both manipulations reduced the TOJ threshold by the same amount. This means that the parameter predicting accuracy on the dichotic TOJ task is the time separating the onset of the first tone from the onset of the second tone (stimulus-onset-asynchrony, SOA, illustrated at the top of Fig 1), regardless of whether that time interval is filled with stimulus stimulation (tone) or silence.

Our finding, however, is limited to the range of stimulus durations we tested, namely tone durations of 10–40 ms. It is unclear whether this finding would extend to other tone durations, since sound duration affects our auditory perception in several ways. First, sound duration affects the loudness of a sound via temporal summation, with sounds being perceived as softer or louder when duration is decreased or increased (respectively), up to 200 msec [28]. Second, sound duration affects our ability to perceive pitch, with lower frequencies requiring longer sound durations than higher frequencies. Third, sound duration also affects our ability to localize a sound source, with longer sounds being localized better by allowing the listeners to move their head towards the sound source [28, 29].

The design of our study directed us toward testing tone durations shorter than those we used in our earlier study. The dichotic TOJ ISI threshold was found to be around 60 msec in several studies [1–3, 11, 12, 14, 17, 20, 23], therefore, manipulating tone duration, ISI and SOA necessarily places an upper limit on the tone durations one can test, i.e., they must be shorter than 60 msec. This means that in order to expand the range of tone durations necessary to test the generalization of our ISI-tone duration TOJ equivalence hypothesis, we focus on shorter durations than those we used in the previous study [1] (i.e., less than 10 msec). Such short

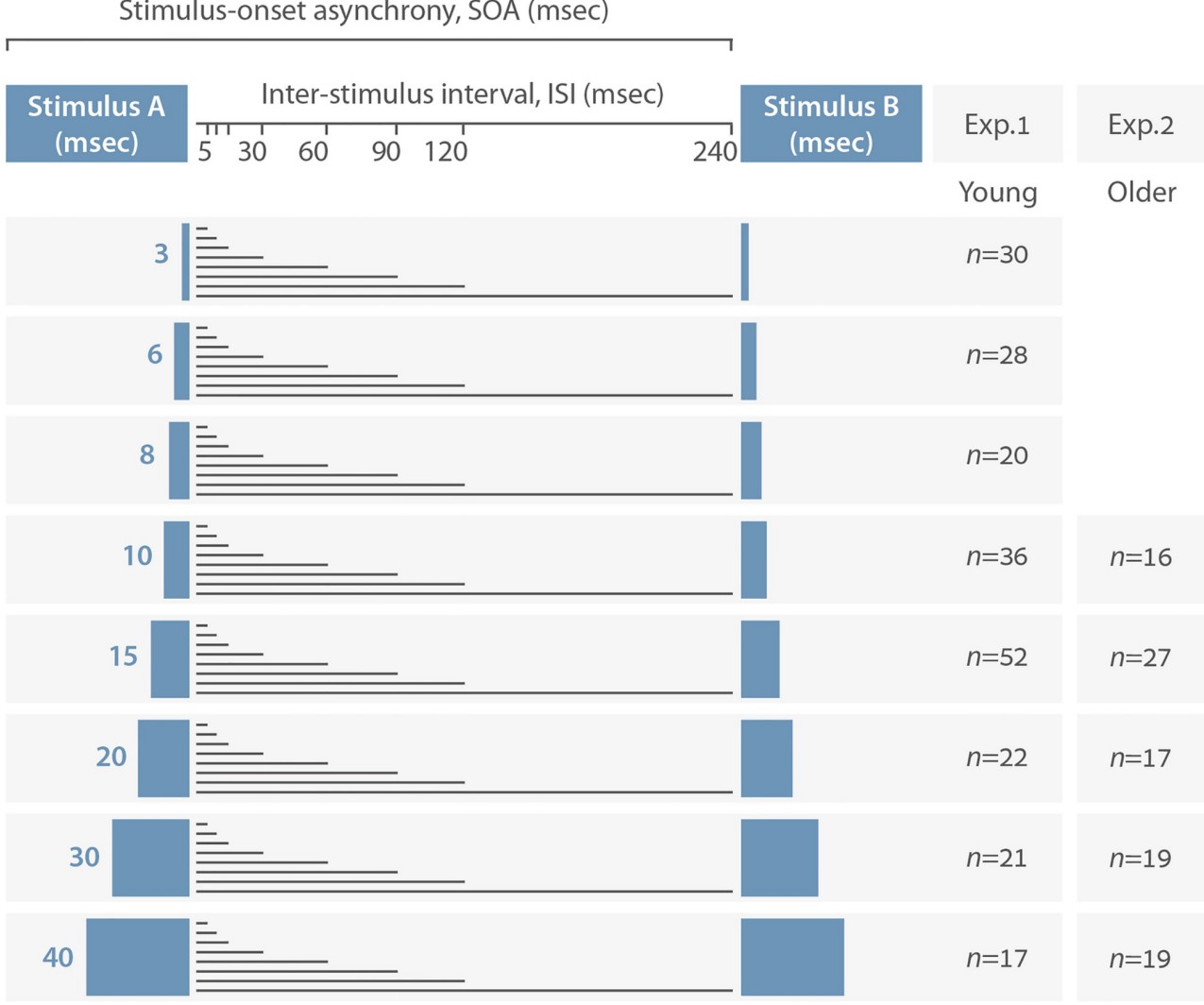

**Fig 1. Schematic illustration of study design demonstrating the relationship of stimulus duration, inter-stimulus interval (ISI), and stimulus-onset asynchrony (SOA) (see top line).** The manipulation of stimulus duration is presented as the duration of Stimulus A and Stimulus B, which varied across groups in the current study as a between-subjects variable. ISI is presented as the silent gap between the offset of Stimulus A and the onset of Stimulus B, which varies within each group as a within-subjects variable. Numbers of participants in each group for Experiments 1 and 2 are shown. **a**. Data by duration and ISI. **b**. Data by SOA.

durations can create transients (short-duration sounds with high amplitude that can accompany the beginning of short sounds) that spread energy across the frequency range [30–33], possibly resulting in different ISI-duration patterns than those observed with tone durations longer than 10 msec. Therefore, in the present study we aimed to test whether our finding applies to very short tone durations (i.e., 3, 6, and 8 msec) as well as durations in the 10–40 msec range, while using the same dichotic TOJ design as Babkoff & Fostick [1].

Furthermore, data from our previous study [1] were robust in showing that, among young adults, a constant time period between the onsets of the two tones (SOA) elicited accurate identification of their order, regardless of the durations of the sound (tone) and silence (ISI). Therefore, as the sum total of the duration and ISI (i.e., SOA) was constant, young adults appear to extract the same temporal information from both elements. It is not yet clear

whether the SOA is also the main predictor for dichotic TOJ performance in other populations, e.g., older adults, who may differ in their response to stimulus duration (*stimulation*) and/or ISI (*lack of stimulation*). Indeed, older adults have difficulty processing short and rapid stimuli. This difficulty is often reflected in the difficulty of older adults in perceiving speech, especially when the speaker talks fast or when speech is accompanied by background noise [3, 20, 21, 34–37]. Studies of temporal processing among older adults, including the studies using speech stimuli, have reported deficiencies in their performance compared to young adults [5, 7, 10, 20, 38–45]. These studies demonstrated that older adults required longer sound durations [38, 39, 43], ISIs [5, 7, 10, 20, 41], and longer gaps within sounds [40, 42, 44, 45], than young adults, in order to correctly perceive them. Such findings suggest that older adults might be sensitive both to tone duration and ISI.

However, since no study has directly measured the mutual contribution of these two variables to the TOJ threshold in older adults, follow-up questions arise: Do older adults extract the same temporal information from the stimulus duration as from the ISI, so that each of these variables has the same effect on their dichotic TOJ threshold, as is the case for younger adults? Or do older adults extract different temporal information from the stimulus duration than from the ISI, so that each of these variables has different effects on their resulting dichotic TOJ threshold? If the pattern of TOJ performance by older adults is similar to that of younger adults, a "zero" line slope would be expected when thresholds are plotted as a function of SOA. This "zero" line slope is expected for both populations, although the thresholds for older adults are expected to be longer than that of younger adults due to age-related temporal deficits among older adults. However, if older adults have greater difficulty processing shorter duration than longer duration tones, the slope of the line relating the dichotic TOJ threshold to tone duration should be significantly greater than "zero", indicating a greater contribution to dichotic TOJ threshold of tone duration than just the increase in SOA. To address these questions, in the current study we repeated the dichotic TOJ study using the same design as Babkoff & Fostick [1] but with participants whose ages ranged from 60–75 years.

The aim of the present study was, therefore, to test our previous conclusion that dichotic TOJ is determined by stimulus onset asynchrony (SOA)—the time separating the onset of the first tone from the onset of the second one [1]—a) by using stimulus parameters that test its boundaries, and b) to determine if it can be generalized. We operationalized this aim by implementing two different manipulations to our previous research model: a) extending the range of tone durations to include also very short tone durations (3–8 msec) thus testing TOJ thresholds with tone durations ranging from 3–40 msec (Experiment 1), and b) using the same experimental methodology as in the previous study [1] to test a population of older adults (Experiment 2). As depicted in Fig 1, which illustrates the current study design, stimulus duration was manipulated across eight groups in Experiment 1 and across five groups in Experiment 2, each of which was presented with a different tone duration; within each group, participants were requested to judge the order of the tones while the ISI was manipulated.

## Experiment 1: Young adults, stimulus duration 3 to 40 msec

### Materials and method

**Participants.** Participants were 226 undergraduate students (136 females, 90 males), aged 20–35 years (mean = 25.5, *SD* = 2.8) who volunteered to participate in the study. The current analyses include participant data presented in the earlier paper (*n* = 65) [1] together with the data from an additional 161 participants (current study). All participants were screened for normal hearing (thresholds of 20 dB HL or less at 500, 1,000, 2,000, and 4,000 Hz, an inclusion criterion). Diagnosis of a learning disability or attention deficit hyperactivity disorder were

exclusion criteria. Participants were divided into eight groups, each of which was tested with only one tone duration, as follows: 3 msec ($n$ = 30), 6 msec ($n$ = 28), 8 msec ($n$ = 20), 10 msec ($n$ = 36), 15 msec ($n$ = 52), 20 msec ($n$ = 22), 30 msec ($n$ = 21), and 40 msec ($n$ = 17).

*Task and stimuli.* We used the experimental design reported in Babkoff and Fostick [1]. In short, participants were presented with two 1 kHz pure tones at a level of 40 dB SL. The tones were presented asynchronously to the right and left ear, and participants were asked to report the order in which they heard them (either right-left or left-right). The tone duration for each participant was 3, 6, 8, 10, 15, 20, 30, or 40 msec, according to their assigned group (between subjects design). Rise/fall times were 1 msec. Eight different ISIs of 5, 10, 15, 30, 60, 90, 120, and 240 msec were randomly used. Each ISI value was repeated 16 times, producing 256 trials (8 ISIs × 2 presentation orders × 16 repetitions). After every 32 trials, participants received a short break. Dichotic TOJ thresholds were defined as the ISI necessary for 75% accuracy, estimated using the best linear approximation of a psychometric function.

The experiment was preceded by a training session performed with tones of the same duration as in the experiment. This was designed to familiarize participants with the sounds and to ascertain whether they correctly reported the ear that was being presented with the sound (right or left) [see 1]. Participants received feedback for their responses during training sessions, but no feedback was presented during the experiment.

**Apparatus.** The hearing screening test was performed using a Danplex DA64 audiometer. The experiment was performed on a Dell laptop computer and the sounds were delivered through TDH-49 headphones.

**Procedure.** The study was approved by the Ariel University Institutional Review Board, and prior to the experiment participants provided written informed consent. Participants were screened for normal hearing prior to the experiment, after signed informed consent was obtained. In addition, their absolute threshold for 1 kHz was measured using the same computer and headphones that were used in the study. The experiment, including the screening and training procedures, took 30–45 minutes.

## Results

**Accuracy.** The accuracy data were transformed by probit (transformation for linearizing sigmoid distributions of proportions [46]. Psychometric functions of the probit-transformed data for the proportion of 'left leading' responses, as a function of ISI, are presented in Fig 2a, separately for each of the eight stimulus durations. A two-way repeated measures analysis of variance (ANOVA) was performed with the probit-transformed data as the dependent variable, ISI as a within-subjects variable, and Stimulus Duration as a between-subjects variable. The analysis revealed main effects of both ISI [$F(7,1358)$ = 948.142, $p < .001$, ηp2 = .830] and Stimulus Duration [$F(7,194)$ = 6.456, $p < .001$, ηp2 = .189], as well as an ISI × Stimulus Duration interaction [$F(49,1358)$ = 1.670, $p = .003$, ηp2 = .057]. Post-hoc ANOVAs between Stimulus Duration for each ISI revealed significant effects of stimulus duration at the short ISIs [5 msec: $F(7,194)$ = 4.288, $p < .001$; 10 msec: $F(7,194)$ = 5.038, $p < .001$; 15msec: $F(7,194)$ = 9.599, $p < .001$; 30 msec: $F(7,194)$ = 5.573, $p < .001$], but not at the longer ISIs (60, 90, 120, and 240 msec; $ps > .05$).

The point of subjective equivalence (PSE) for left-leading responses of 50% was calculated on the probit-transformed data using a linear equation. The PSE for all eight stimulus durations was 0 msec (-6.25E-15 to 6.38E-15 msec). Fig 2b presents a scattergram of the same probit-transformed accuracy data when tone duration and ISI were incorporated into one measure, namely, the SOA (the total delay between tone onset at the leading ear and tone onset at the lagging ear). For the transformed SOA data, the PSE was 0 msec, similar to the PSE obtained from the ISI data for all tone durations. The linear component for the data, plotted as

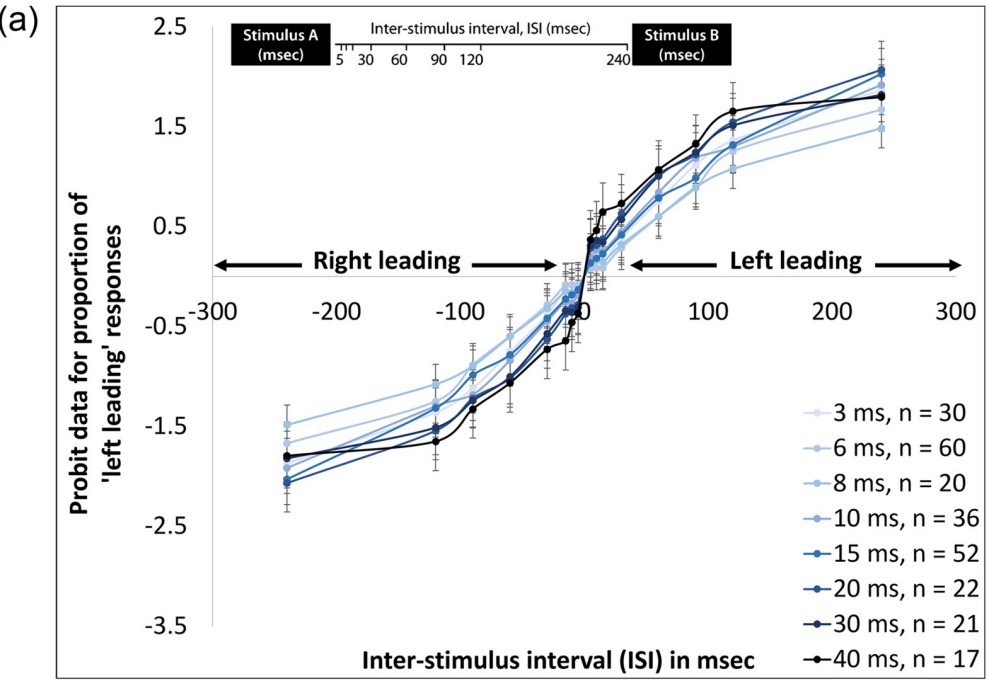

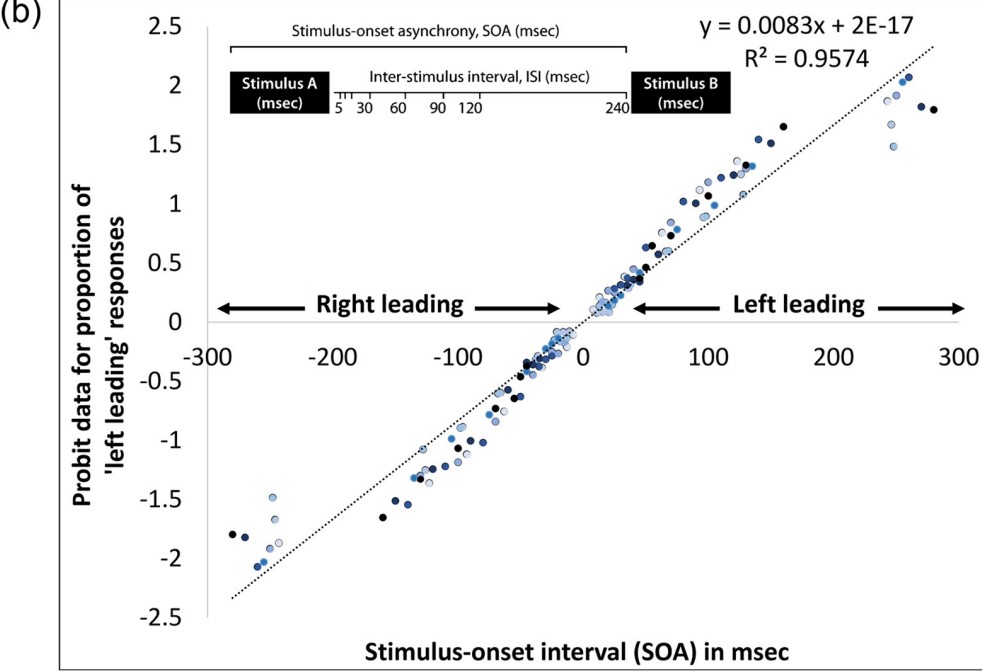

**Fig 2. Psychometric function of probit-transformed data for the proportion of 'left leading' responses of young adult participants across eight different stimulus durations for: (a) each stimulus duration by ISI; and (b) all data by SOA.** Schematic diagrams illustrating the distinction between ISI and SOA appear in each panel. **a**. Data by duration and ISI. **b**. Data by SOA.

a function of the SOA, predicted 95.7% of the variance in accuracy. Notwithstanding, the points below and above SOAs of -200 and +200 msec were out of line with the rest of the data. This might be due to an asymptotic performance at the longest ISI values. Repeating the analysis without these values included resulted in a predictive value of 98.9% (y = 0.0103x − 6E-18).

**Thresholds.** TOJ ISI thresholds, defined as the ISI necessary for 75% accuracy, were estimated using a linear function. ISI thresholds are presented in Fig 3a as a scattergram for all participants plotted against each tone duration. These data ranged correspondingly between 58.4–25.4 msec, with stable standard errors in the range of 5.3–7 msec. Heterogeneity testing (Levene Statistic) was not significant ($F(7,316) = 1.842$, $p = .079$). Group mean data are also plotted (Fig 3a) and were tested against a model that predicted a reduction in threshold for the same magnitude of increase in tone duration. The data were found not to deviate significantly from this model (probit analysis, $Z = -3.13$; $p = .002$). The best linear fit to the mean ISI thresholds ($R2 = 0.69$, $p < .001$) is depicted in Fig 3a (see straight line) as is the predicted line (based on $y = a−bx$, predicting a similar reduction in ISI threshold as the increase in tone duration; see dotted line). The slope of the predicted line related to the current study's tone duration range of 3–40 msec was -1.17, which does not differ significantly from the slope of -0.86 reported for the more limited 10–40 msec range of tone durations from our previous study [1] ($t(290) = 1.162$, $p = 0.123$). Furthermore, the Bayes factor ($BF_{01} = 3.49$) suggested substantial support for $H_0$. This indicates that the current data are more likely to occur under $H_0$, namely, there was no difference between the slopes produced by the data in the current and the previous study [1].

In Fig 3b, dichotic TOJ thresholds are plotted in terms of the SOA, as a function of tone duration. Note, the scattergram data and the averages fall very close to, or on, the zero-slope dotted line. The point at which the average dichotic TOJ threshold (SOA) crosses the vertical axis in Fig 3b is 69.79 ± 10.45 msec (probit analysis, $Z = 5.04$, $p < .001$).

## Discussion

Extending the range of tone duration beyond 10–40 msec to include shorter durations of 3–8 msec did not change the general pattern of a "tradeoff" between tone duration and ISI [1]. When the data from all 226 participants were analyzed, there was a decrease of 1.17 msec in ISI for every increase of 1 msec in tone duration over the whole range of 3–40 msec. This was not different from the previous study of 65 participants using a tone duration range of 10–40 msec, in which there was an observed decrease of 0.86 msec in ISI for each increase of 1 msec in tone duration.

The data from Experiment 1 show that, for stimulus durations of 3–40 msec, young adults utilize the same cue for temporal processing from both the stimuli (tone duration) and from the silent gap between them (ISI). The extension of the range of the tone duration from 10–40 msec to 3–40 msec in the present study supports our earlier conclusion that the primary predictor in judging dichotic temporal order is the temporal lag between the onsets of the two stimuli, namely, the stimulus onset asynchrony (SOA) [1]. Therefore, among young adults and within the tone duration range of 3–40 msec, dichotic TOJ thresholds (measured as the SOA) are invariant to tone duration. Moreover, as the average dichotic TOJ threshold crosses the vertical axis at approximately 57 msec (as found in our previous study [1]) to 70 msec (as in the current study, Fig 3a and 3b), this suggests that the time between the onset of two tones that is required for perceiving their order is constant (around 60–70 msec). Future studies should extend the investigation to variations of other parameters of the presented TOJ sound stimuli, such as spectrum and intensity, to further explore the boundaries of this time constant.

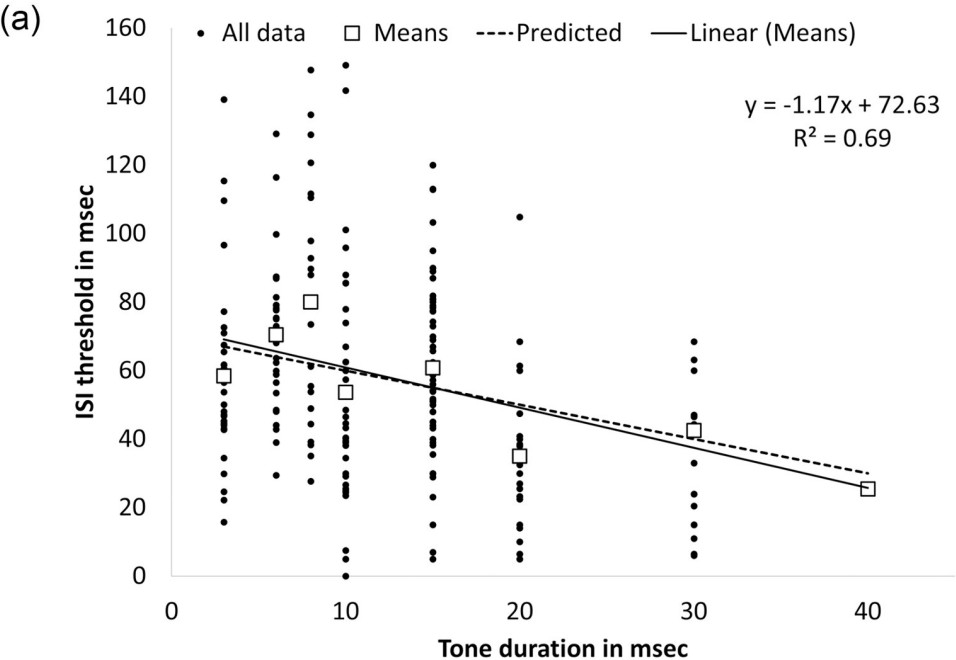

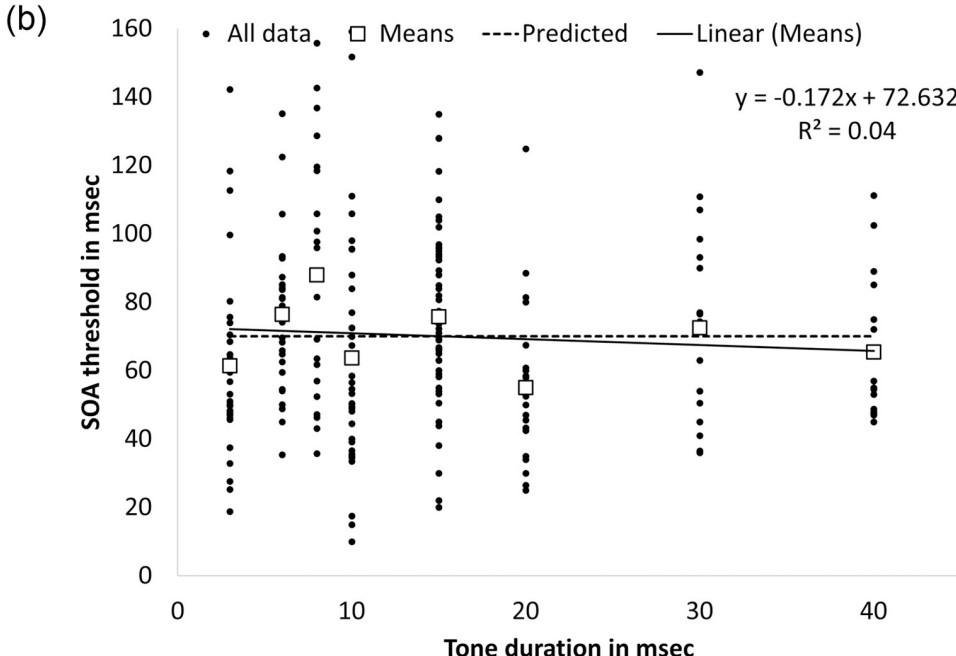

**Fig 3. TOJ thresholds of young adult participants across eight different stimulus durations for: (a) ISI thresholds (silent gap between the offset of the stimulus at the leading ear and the onset of the stimulus at the lagging ear); and (b) SOA thresholds (duration of the stimulus at the leading ear added to the ISI threshold). a**. ISI thresholds. **b**. SOA thresholds.

## Experiment 2: Older adults, stimulus duration 10 to 40 ms

### Materials and method

Experiment 2 was conducted using the same methodology as Experiment 1, with the exception of: a) older participants and b) a duration range of 10–40 msec. A group of 98 participants (59 females, 39 males), aged 60–75 years (mean = 66.4, $SD$ = 6.1), volunteered to participate in the study. Participants were divided into five groups, each of which was tested with only one tone duration, as follows: 10 msec ($n$ = 16), 15 msec ($n$ = 27), 20 msec ($n$ = 17), 30 msec ($n$ = 19), and 40 msec ($n$ = 19). After providing signed informed consent, the participants were screened for age-normal hearing (hearing thresholds of 35 dB HL or less at 500, 1,000, 2,000, and 4,000 Hz). This was an inclusion criterion while hearing deficit was an exclusion criterion.

### Results

**Accuracy.**  The accuracy data were transformed by probit (transformation for linearizing sigmoid distributions of proportions [46]. The psychometric functions of the probit-transformed data for the proportion of 'left leading' responses, as a function of ISI, are presented in Fig 4a, for each of the four stimulus durations. A two-way repeated measures ANOVA was performed with the probit-transformed data as the dependent variable, ISI as a within-subjects variable, and Stimulus Duration as a between-subjects variable. The analysis revealed main effects for the ISI [$F_{(7,651)}$ = 244.105, $p < .001$, ηp2 = .724] and an ISI × Stimulus Duration interaction [$F_{(28,651)}$ = 3.291, $p < .001$, ηp2 = .124], but no main effect for the Stimulus Duration [$F_{(4,93)}$ = 1.165, $p = .332$, ηp2 = .048]. Post-hoc ANOVAs of Stimulus Duration for each ISI revealed significant effects of stimulus duration at some ISIs [5 msec: $F_{(4,93)}$ = 5.647, $p < .001$; 15msec: $F_{(4,93)}$ = 6.879, $p < .001$; 240 msec: $F_{(4,93)}$ = 4.714, $p = .002$], but not most (10, 30, 60, 90, and 120 msec; $ps > .05$).

The PSE for 'left-leading' responses of 50% was calculated on the probit-transformed data using a linear equation. The PSE for all eight stimulus durations was 0 msec (-3.80E-15 to 1.39E-15 ms). Fig 4b presents a scattergram of the same probit-transformed accuracy data when tone duration and ISI were combined as the SOA. The linear component for the data, plotted as a function of the SOA, predicted 92.9% of the variance in accuracy.

**Thresholds.**  TOJ ISI thresholds (i.e., ISI required for 75% correct responses) for the older adult cohort are plotted as a scattergram as a function of tone duration in Fig 5a. Mean ISI thresholds ranged between 85.2–63.2 msec for stimuli durations of 10–40 msec, with standard errors in the range of 5.5–7.2 msec. Group mean thresholds are also plotted. We tested the group mean data against the predicted model and found that the data do not deviate significantly from this model (probit analysis, $Z$ = -5.989; $p < .001$). The best linear fit to the means ($R2$ = 0.91, $p < .001$) is depicted in Fig 5a (see straight line) as is the predicted line (based on $y = a—bx$ (i.e., dotted line). The slope of the predicted line (-0.69) was not significantly different from the slope found previously for the same stimulus duration among young adults (-0.86; $t_{(162)}$ = -0.264, $p = 0.396$), and not different from the slope found in Experiment 1 for stimulus durations of 3–40 msec among young adults (-1.17; $t_{(322)}$ = -1.274, $p = 0.102$). The Bayes factor for the current data compared both to our previous study ($BF_{01}$ = 5.64) and to the data from Experiment 1 ($BF_{01}$ = 3.48) suggested substantial support for the $H_0$. This indicates that the current data are more likely to occur under $H_0$, namely, there was no difference between slopes of the current study when compared to the previous study [1] and to Experiment 1.

In Fig 5b, dichotic TOJ thresholds (SOAs) are plotted against tone duration. Note, the scattergram data and the averages fall very close to-, or on, the zero-slope dotted line. The point at

(a)

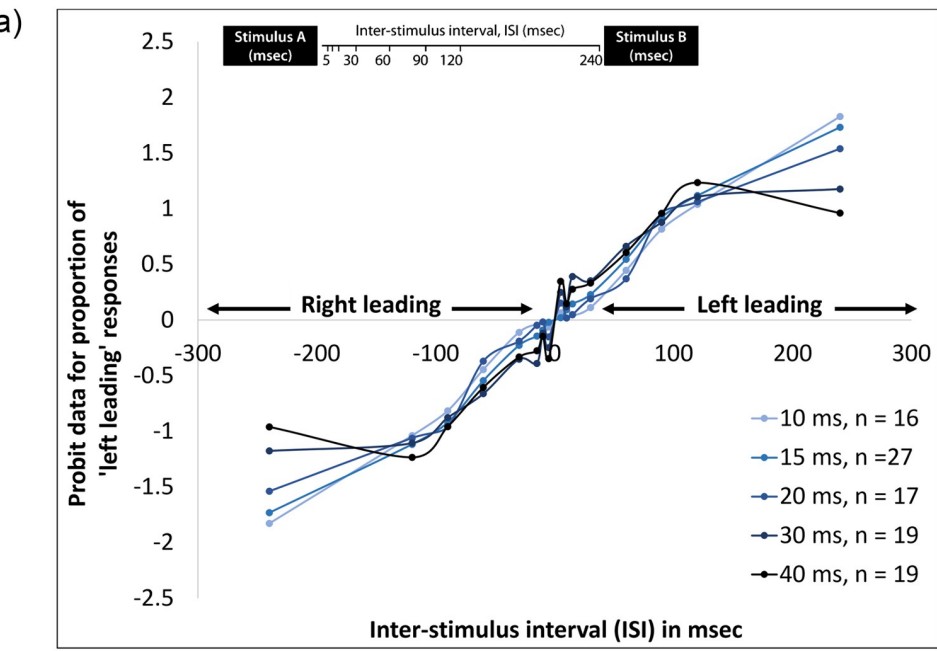

(b)

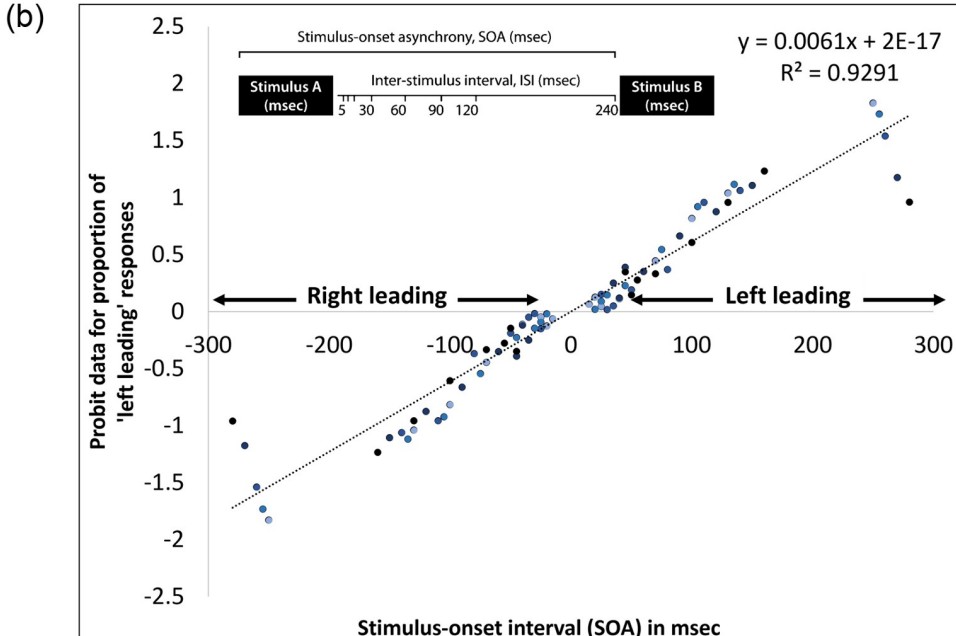

**Fig 4. Psychometric function of probit-transformed data for the proportion of 'left leading' responses of older participants across five different stimulus durations for: (a) each stimulus duration by ISI; and (b) all data by SOA.** Schematic diagrams illustrating the distinction between ISI and SOA appear in each panel. **a**. Data by duration and ISI. **b**. Data by SOA.

which the average dichotic TOJ threshold (SOA) crosses the vertical axis in Fig 5b is 100 msec (probit analysis, $Z = -2.74$, $p = .02$), indicating that the TOJ threshold for the older adults is, on average, 33 msec longer than the young adults in Experiment 1 (99.6 msec vs. 66.5 msec, respectively).

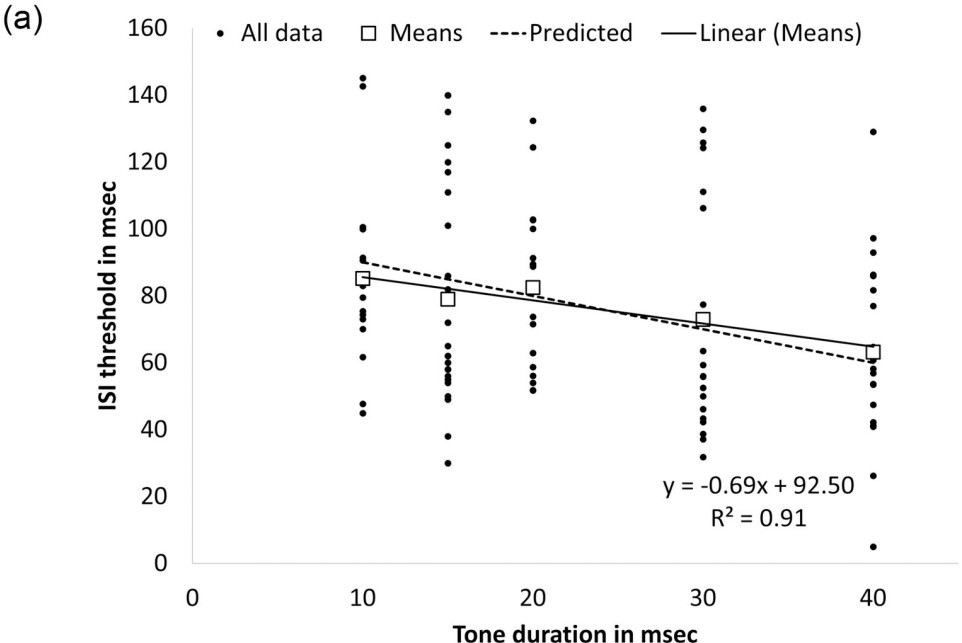

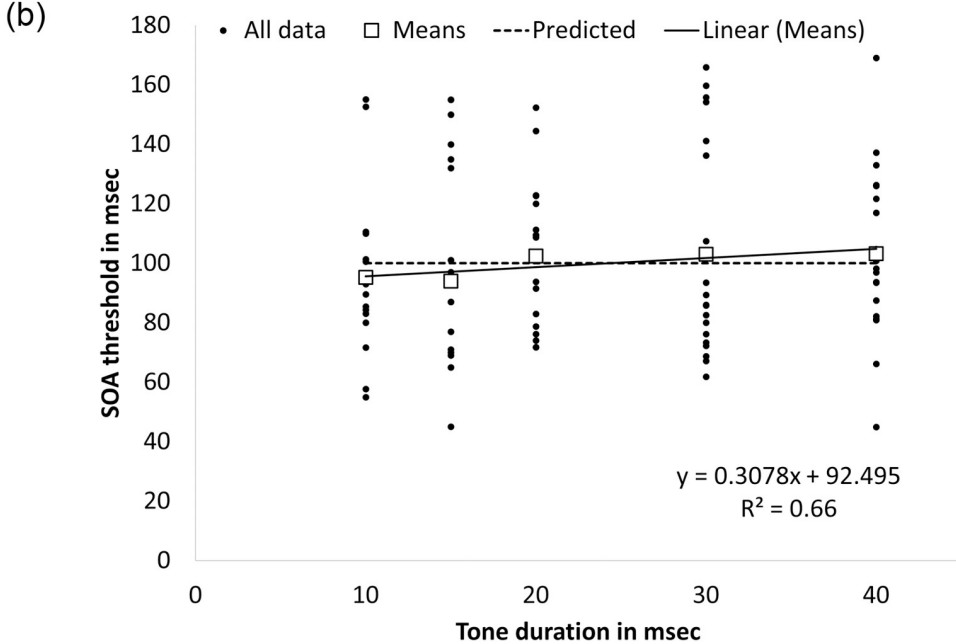

**Fig 5. TOJ thresholds of older participants plotted as a function of stimulus duration for (a) ISI thresholds (silent gap between the offset of the stimulus at the leading ear and the onset of the stimulus at the lagging ear); and (b) SOA thresholds (duration of the stimulus at the leading ear added to the ISI threshold).** a. ISI thresholds. b. SOA thresholds.

## Discussion

Extending our study to an older adult population did not change the general pattern of a "tradeoff" between tone duration and ISI in predicting dichotic TOJ thresholds, as found among young adults in our earlier study [1] and in Experiment 1. As predicted, the thresholds

for older adults were longer than for the younger adults; on average, the older adults' TOJ threshold was longer by 33 msec across a 10–40 msec range of stimulus durations (i.e., an average SOA of 99.6 msec for the older adults vs. 66.5 msec for the younger adults). Moreover, among the older adults, the results indicated a decrease of 0.69 msec in ISI for every increase of 1 msec in tone duration from 10–40 msec, which does not differ significantly from the 0.86 msec decrease observed among younger adults in the previous study [1] over the same tone duration range.

The purpose of Experiment 2 was to validate the conclusion reached in our earlier study [1] —that the SOA is the main parameter predicting accurate judgment of temporal order of two tones presented dichotically—this time among an older adult cohort. The current data indicate that the SOA is indeed the major parameter predicting dichotic TOJ performance when testing among older adults, who typically show a general deficit in temporal processing (e.g., longer TOJ thresholds). While older adults do need more time than younger adults between the onset of first dichotic tone in the leading ear to the onset of the second dichotic tone in the lagging ear in order to judge the order correctly, the general relationship between dichotic TOJ ISI threshold, and tone duration is the same as for the younger adults. It is the SOA, rather than the silent interval (ISI) or the tone duration, that is the crucial parameter utilized by both younger and older adults in judging temporal order.

## Summary and conclusions

The present study aimed to test the generalizability of our previous finding [1] that dichotic TOJ performance is best predicted by stimulus onset asynchrony (SOA), namely, the time separating the onset of the first tone to the onset of the second one. We did so by implementing two different manipulations to our previous research model: 1) extending the range of tone durations to also include 3–8 msec among a population of young adults; and 2) testing the ISI-tone duration relationship among a cohort of older adults, among whom general deficits in auditory temporal order judgment have been shown in prior research [2, 3, 5, 7, 10, 20, 22]. One might have expected to find a larger contribution to TOJ performance of tone duration because tones of 10 msec, or shorter, spread energy across the frequency range. This may change the *quality* of the tones, making the impact of the duration itself on TOJ threshold greater through the effect on tone quality, and not only through its impact on the temporal separation between the onsets of the first and second tones. Moreover, as older populations have shown previous deficits in dichotic TOJ [i.e., evidencing longer TOJ ISI thresholds] one might also have expected to observe a non-linear relationship between tone duration and TOJ ISI threshold among our older adult cohort, because they may need more time to perceive and process temporal information. However, the results of both experiments seem to support the hypothesis that the dichotic TOJ threshold is determined by a *general* temporal mechanism, whether the tone duration is as short as 3 msec or as long as 40 msec, and regardless of older adults' general deficits in temporal processing. This indicates that the temporal mechanism for dichotic TOJ is affected by temporal asynchrony, independent of the *nature* of that asynchrony (whether filled with silence or sound), as long as the gap between tones sufficiently conveys the information of asynchrony.

The present and the previous study [1] were both conducted utilizing auditory dichotic TOJ (also referred to as spatial or binaural TOJ). This TOJ paradigm involves uses two identical sounds presented asynchronously to the right and left ears; other auditory TOJ paradigms use two tones that differ in pitch or spectrum and are presented either monaurally or diotically (to both ears at the same time). The advantage of measuring temporal processing using dichotic TOJ is that the stimuli are identical, providing assurance that the temporal judgment is based on the temporal relationship of the two stimuli alone and not on other cues such as

pitch [2, 3, 19]. Furthermore, perception of the stimulation of both ears by two asynchronous sounds, by definition, reflects mainly central auditory processing [1, 11, 47]. Consequently, the conclusions drawn from the current and earlier studies are limited to TOJ as tested by this paradigm, which has been shown to mainly involve temporal cues [2, 3, 11, 14, 17]. The extent to which these conclusions can be generalized to other TOJ paradigms has yet to be tested.

Several theoretical and clinical conclusions arise from the experiments carried out in the present study. Our first conclusion is that when judging the order of two tones presented to the two ears, individuals extract the same temporal information from the stimuli as from the silent gap between them, whether they are young and have intact temporal processing abilities or are older and have less than intact temporal processing abilities. Second, the same temporal parameter–SOA–impacts the absolute value of dichotic TOJ threshold regardless of whether an individual has good or poor temporal processing ability; this dichotic TOJ threshold will then remain constant across different tone durations. Thus, temporal processing abilities affect the dichotic TOJ mechanism only *quantitatively*, rather than qualitatively. Clinically, this conclusion is important since quantitative changes can be more easily remedied than those that are qualitative. We recommend future research to further refine the conclusions of the current study by conducting dichotic TOJ testing using sounds of different intensities and spectra, employing additional temporal processing tasks, and studying diverse populations with varied temporal processing abilities.

## Author Contributions

**Conceptualization:** Harvey Babkoff.

**Data curation:** Leah Fostick.

**Formal analysis:** Leah Fostick.

**Methodology:** Leah Fostick, Harvey Babkoff.

**Project administration:** Leah Fostick.

**Supervision:** Harvey Babkoff.

**Validation:** Harvey Babkoff.

**Writing – original draft:** Leah Fostick.

**Writing – review & editing:** Leah Fostick, Harvey Babkoff.

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
