## [Decision Letter · Decision Letter 0]

13 Oct 2021

PONE-D-21-18616

The role of tone duration in dichotic temporal order judgment II: Extending the boundaries of duration and age

PLOS ONE

Dear Dr. Fostick,

Thank you for submitting your manuscript to PLOS ONE. I now have reviews from two individuals who are truly experts in the field. I always feel fortunate when it is possible to get input from such highly qualified authorities. But as you will see, the conclusions reached concerning your manuscript were divided across the reviewers, with the first reviewer decidedly more positive than the second. I believe that all of the observations offered by both reviewers are well justified - and addressable. Many of these  concerns have to do with presentation of the experiment and results, rather than with the conduct of the experiment itself. Therefore, I would like to invite you to submit a revision of your manuscript after you address all their concerns.

We look forward to receiving your revised manuscript.

Kind regards,

Susan Nittrouer, Ph.D.

Academic Editor

PLOS ONE

2. In order to improve reporting, in your methods section, please provide additional information about the participant recruitment method and the demographic details of your participants, such as table of relevant demographic details.

Additional Editor Comments (if provided):

Reviewers' comments:

Reviewer's Responses to Questions

**Comments to the Author**

1. Is the manuscript technically sound, and do the data support the conclusions?

Reviewer #1: Partly

Reviewer #2: Yes

2. Has the statistical analysis been performed appropriately and rigorously? 

Reviewer #1: No

Reviewer #2: Yes

3. Have the authors made all data underlying the findings in their manuscript fully available?

Reviewer #1: No

Reviewer #2: Yes

4. Is the manuscript presented in an intelligible fashion and written in standard English?

Reviewer #1: Yes

Reviewer #2: No

5. Review Comments to the Author

Reviewer #1: This is a well-written and concise manuscript addressing temporal order judgments, following on the authors’ previous work.

Background and motivation for the study: It is an expanded study of tone duration in temporal order judgment (TOJ). The study contains a large group of young adults (n=226) and older adults (n = 98). It Extends previous work by including shorter stimulus durations and by including older listeners.

Stimulus-onset-asynchrony (SOA) continues to explain the TOJ even for shorter durations and abilities. TOJ generally changes with age. (The differences between age groups are Quantitative, NOT qualitative)

The authors conclude that tone duration, then, just provides more information about SOA. It seems to fit the idea that stimulus onsets are very important for perception. A question of significance and motivation for the study arises: is this new information that contributes significantly to what we know?

In asking that, it seems that the motivation for the use of shorter duration stimuli isn’t strong. I’m not sure why we expected that the results from 3-ms stimuli would be different from 10-ms stimuli (and from Fig 1 perhaps they are). If the stimuli are appropriately ramped, then spectral splatter should be minimized and one might expect this result. What were the temporal ramps applied to the stimuli? Also, based on Figure 1a. it appears that the 3-ms stimuli do separate themselves from the other data, with flatter functions.

The reference for spectral splatter of short duration tones is quite out-dated (1967) and presumably instrumentation has changed dramatically since that time. Suied et al from JASA in 2014 have more recent data, and there are probably others as well.

It does seem interesting to address the question of the aging auditory system. Perhaps one might hypothesize that the only differences between younger and older listeners in TOJ would be found with short duration stimuli. This is introduced on page 10. However, then the older adults were tested using 10 to 40 ms stimuli, so it’s challenging to connect experiments 1 and 2. It does not appear that the age-related differences are duration specific, and that seems like an important point that could be emphasized more strongly. I think that question could be addressed more directly in the figures and in the analysis.

I have questions about the fit in Figure 2. I believe readers will need more information about that. Visual inspection shows a great deal of variability and thus it is difficult to understand the proportion of variance that is explained by the linear fit. Was heterogeneity tested? More details are need for me, and presumably other readers, to understand the data.

Fig 5. Is there a relationship between age and threshold among the older listeners? Age could be considered as a continuous variable.

In the Discussion section the authors state: as predicted the older adults’ thresholds were 33 ms longer than younger? Where was this predicted?

Data from younger and older listeners do appear from these data to be qualitatively similar, but it’s not clear enough yet why this is important. It is also unclear how experiment 1 fits into the overall picture. Additional motivation is needed, and details are needed about the stimulus characteristics (ramps) and the statistical analyses.

Reviewer #2: Title: The role of tone duration in dichotic temporal order judgment II: Extending the

boundaries of duration and age

Authors: Leah Fostick and Harvey Babkoff

Submitted to: PLOS ONE

Manuscript number: PONE-D-21-18616

Overview

The idea proposed in this manuscript it that the ability to determine the temporal order of two auditory stimuli is determined by the time between the onsets of the two stimuli (stimulus onset asynchrony, SOA) rather than by the time between the offset of the first stimulus and the onset of the second stimulus (inter stimulus interval, ISI). The authors test this proposal by measuring the proportion correct determinations of temporal order for tones of different durations at each of multiple ISIs, and then comparing how the TOJ threshold is affected by tone duration when the TOJ threshold is based on the ISI versus when it is based on the SOA. The ISI-derived thresholds decrease with increasing tone duration, while the SOA-derived thresholds are constant across tone duration, suggesting that the SOA is the critical cue for performance. The same general pattern is reported for tone durations ranging from 3 to 40 ms in young adults (an expansion from 10 to 40 ms reported in a previous paper by the same authors), and for tone durations ranging from 10 to 40 ms in older adults.

I provide my general comments, and then my more specific comments on the manuscript, below.

General Comments

Stimulus description: I think it is essential to include a description of how the threshold is determined (see other general comment), to include a figure illustrating how manipulating the tone duration allows the determination of whether the dichotic TOJ threshold is determined by the ISI or the SOA, and to include stimulus schematics in each figure to illustrate how the data are being analyzed (based on ISI or SOA). It took me quite a while to understand the stimuli. For example, here is a version of a note to myself as I was reading: Text [Pg. 3, bottom]: “We asked whether changes in the duration of the tones and inter-stimulus interval…affect dichotic temporal order judgment accuracy in the same or different ways” My note: “To me, it is confusing to introduce ISI here, because, I suspect, the TOJ threshold is determined by manipulations of the ISI…Oh!! Is the idea that the ISI is a silent period between the stimuli, but that in the manipulation of stimulus duration, the two tones are always contiguous??” Now I understand, or think I understand, that there are ISIs of various lengths between the two tones for all of the stimuli, that the stimulus duration was varied across conditions (with multiple ISIs for each stimulus duration), and that the question was whether the data are better explained as a whole by evaluating performance based on the ISI or on the stimulus onset asynchrony (SOA). This is a case where a picture really would be worth a thousand words.

Threshold estimate: Why is the estimate of the time required between the onset of the two tones to determine their order—around 67 ms for younger adults--so much longer than the 15-20 ms value reported by Hirsh (1959)…a classic paper on temporal order judgment that is not cited in the current manuscript? Hirsh (and subsequently many others) used two ~500-ms stimuli whose onsets differed but whose offsets were coincident, but if temporal order is determined by stimulus onset asynchrony then it seems that the results of the present experiment and Hirsh’s data should align. Is the difference due to monotic vs. dichotic presentation?

Writing: I found most of the manuscript to be quite difficult to read. The exception was the Summary and Conclusion section. The previous sister paper by Babkoff and Fostick (2013) is much clearer overall, indicating that the authors are capable of producing clear prose and therefore could greatly improve the clarity of the current manuscript.

Introductions to individual experiments: I recommend combining all of the introductions into a single introduction. As it is now, some portions of the introductions to the individual experiments repeat information in the main introduction and other portions provide information that would be quite helpful to include in the main introduction.

Results: I found the results section of the sister paper by Babkoff and Fostick (2013) to be much clearer and more informative than the current results sections. I recommend modeling the current results sections after the earlier paper while still incorporating new additions like the Bayes factor.

Stimulus duration: It would be quite helpful to spell out why the extension of the investigation to stimulus durations beyond 10-40 ms only focused on stimuli in the 3-8 ms range, rather than on a much wider range of stimulus durations. I think the reason is that the dichotic TOJ threshold is around 60 ms, so to compare ISI and SOA requires durations shorter than 60 ms. There is some mention of the possibility that the spectrum of the shortest stimuli would affect the outcome, but I did not find that argument to be compelling, especially without placing the 3-8 ms restriction in the larger context.

TOJ threshold: The TOJ threshold is not defined.

Terminology: The terms temporal order judgment (TOJ) and dichotic TOJ are used interchangeably throughout the manuscript. It would be helpful to select just one term and then stick with it. Is the idea that the dichotic TOJ is just one more example of TOJ or that the dichotic aspect is an important factor? If the focus is on dichotic TOJ, the current claims could be tested with monaural TOJ tests, as well.

Specific Comments

Abstract

Pg. 2, top: “the major predictor of auditory TOJ threshold, and performance on spatial/dichotic TOJ tasks” Is there a difference between the TOJ threshold and performance on TOJ tasks? Is the intent that the predictor is for TOJ tasks in general, and the present results are for dichotic TOJ tasks in particular?

Pg. 2: To me, the abstract as a whole does not capture the major message of the manuscript—that dichotic TOJ thresholds appear to be determined by the SOA, rather than the ISI. I think the abstract would be much stronger if it were introduced using the argument at the beginning of the Summary and Conclusions about how the manuscript provides two tests of the idea that it is the SOA rather than ISI that determines the dichotic TOJ threshold.

Introduction

Pg. 3, top: “Temporal order judgment (TOJ) reflects the individual’s ability to correctly perceive the order of consecutive stimuli presented rapidly.” This sentence is not clear to me. It seems to conflate the task (temporal order judgment) with performance on the task (correctly perceive…and presented rapidly).

Pg. 3, top: “TOJ thresholds” What is a TOJ threshold? I think it would help to introduce the ideas a bit more slowly. For example, I suspect that the TOJ threshold is determined by varying the ISI.

Pg. 3, top: “TOJ thresholds were found to be related to phonological skills [1-10] and to speech perception [2, 8,11-16].” What was the direction of the relationship? I assume that higher thresholds were associated with poorer phonological skills and poorer speech perception, but it would be helpful to state that directly.

Pg. 3, middle: “use stimuli that differ in spectrum or in duration [17], in frequency (pitch)” I do not understand the distinction between spectrum and frequency. The spectra differ for sounds of two different frequencies. I suspect that intent is that ‘spectrum’ means the spectrum of a complex sound, but that is not clear from the text.

Pg. 3, middle: “or in synchronicity, meaning which ear receives the first/second stimulus (2,5,9,17-19; 22-23,24-27]” I do not understand the term ‘synchronicity’ in this context. Would ‘ear of presentation’ work? Does synchronicity mean the same thing as dichotic TOJ?

Pg. 3, bottom: “of the tones” what tones?

Experiment 1: Young Participants, Stimulus Duration 3 to 40 ms

Materials and Method

Pg. 5, bottom: “226 participants” It would be helpful to include the number of males and females. Were there any sex differences in performance?

Pg. 6, top: “The age of the participants ranged from 20 – 35 years.” It would be helpful to add the mean and standard deviation of the ages.

Pg. 6, top: Were the participants compensated for their participation?

Pg. 6, bottom: “ranging between 5-240 msec” It would be helpful to list the ISI values.

Pg. 6, bottom: Were the participants given trial-by-trial feedback?

Pg. 6, bottom: “to ascertain whether they perceived the order of the tones and correctly reported the ear stimulated (right or left)” The distinction between perceiving the order of the tones and correctly reporting the ear stimulated is not clear to me. How can one be done without the other? Is the idea that participants could tell that the two tones were presented in opposite ears, sequentially, but could not indicate which ear was first?

Pg. 6: What was the level of the tones?

Pg. 7, top: How were the tones generated?

Pg. 7 middle: “All participants were screened for hearing difficulties and their absolute threshold for 1 kHz was measured using the same computer and headphones that were used in the study.” Was the screening at 1 kHz separate from the screening for normal hearing mentioned in the ‘Participants’ section?

Results

Pg. 7, bottom: “Mean ISI thresholds” How is the ISI threshold defined?

Pg. 7, bottom: “Group mean data was tested against the predicted model and was found not to deviate significantly from this model (Probit analysis, Z = -3.13; p = .002).” What was the predicted model?

Pg. 7, bottom: “Mean ISI thresholds for the data collected in the present study (3 – 8 ms) and the previous study (10 – 40 ms) ranged between 58.4 – 25.4 ms“ The impression I get from this sentence (and elsewhere) is that all of the new data (n=161) are for 3-8 ms, and all of the previous data (n=65) are for 10-40 ms, but that does not fit with the n provided for each duration separately.

Pg. 7, bottom: “ranged between 58.4—25.4 ms” In which direction?

Pg. 7, bottom: Fig. 1B, the points above the ~-250 ms and +250 ms SOA are out of line with the rest of the data. I think this pattern deserves mention in the results section…presumably arises because of a flattening of performance, asymptotic performance at the longest ISI values. What is the outcome if these values are removed from the line fitting? My sense is that these values lead to underestimation of the ‘true’ slope. Is there a reason for using linear vs. log values?

pg. 8, top “TOJ thresholds” How is the TOJ threshold defined? Is it the same as the dichotic TOJ threshold mentioned later in the same paragraph? Is it the same as the ISI threshold mentioned in the preceding paragraph?

Pg. 8, top: “The best linear fit to the means is drawn as a straight line (R2= 0.69,

p<.001) and the predicted line based on y = a - bx is drawn as a dotted line.” What is the predicted line?

Discussion

Pg. 8, bottom: “for a wide range of stimulus durations (3 – 40 msec)” I do not consider 3-40 ms to be a wide range of stimulus durations. I recommend deleting “for a wide range...” to the end of the sentence, and just stating the outcome.”

Pg. 9, top: “suggest that the time between the onset of two tones that is required for perceiving their order is constant (around 60 – 70 ms)” I think this estimate is quite interesting, and deserves to be emphasized in the paper, discussion, and possibly abstract. However, I think that it would be better to point it out first in the results section (I had to go back to the figure to work out how the value was determined).

Experiment 2: Elderly Participants, Stimulus Duration 10 to 40 ms

Materials and Method

Pg. 9, middle: “elderly” (mentioned several time): I recommend using a different term, possibly “older”

Pg. 9, middle: “Studies of temporal processing among the elderly, including those researching it in the context of speech, have reported deficiencies in performance compared to young adults [8,19-20,22,24,26,38-45]. These studies have demonstrated that aging adults require longer sound durations, ISIs, and gaps, than do young adults, in order to correctly perceive them.” It would be helpful to indicate which of the references in the first sentence are associated with each of the measures listed in the second sentence.

Pg. 9, middle: “Such findings support the notion of aging adults’ sensitivity both to tone duration and ISI” I do not understand this phrase. Is the intent that such findings indicate that aging adults are sensitive to both…??

Pg. 10, top: “If so, we expected the pattern of older adults’ performance to be similar to those of young adults, indicating that although the thresholds will be longer, on average, there will be a “zero” line slope when thresholds are plotted as a function of SOA.”�”If so, the pattern of older adults’ performance should be similar to that of young adults, such that there should be a “zero” line slope when thresholds are plotted as a function of SOA.” I recommend stating in a separate sentence why the thresholds are expected to be longer.

Pg. 10, middle: “than longer ones, we would expect the line’s slope”�”than longer ones, the line’s slope (relating dichotic TOJ threshold to tone duration) should be significantly greater than “zero”

Pg. 10 bottom to pg. 11 top: I recommend taking out the subheads under Materials and Methods and just stating that Experiment 2 was the same as Experiment 1, except…(fill in the blanks).

Pg. 11 top: What does ‘resembled’ mean in this context? Were there a number of differences in the procedure between Experiment 1 and Experiment 2, but a vague similarity between the two experiments? I suspect not.

Discussion

Pg. 12, middle: “the thresholds for aging adults were, on average, 33 ms longer than for the younger adults (an average SOA threshold of 99.6 ms for aging adults vs. 66.5 ms for the younger).” See comment above (pg. 9, top) about the estimate of the threshold. I recommend including that information first in the results section.

Summary and conclusions

Pg. 13, middle: “The present study aimed to evaluate the temporal mechanism of TOJ by testing the generalizability of the conclusion that dichotic TOJ is determined by stimulus onset asynchrony (SOA), the time separating the onset of the first tone to the onset of the second one. We tested this hypothesis by two different manipulations: 1) by extending the range of tone durations to include 3-10 msec on a population of young adults; 2) by performing identical testing on a population of older adults, about whom there is existing evidence of a general deficit in auditory temporal order judgment.” I think this set-up is much clearer than the one in the current introduction.

Figure Captions

Figures

After taking the time to write down the suggestions listed below for improving the figures, I finally looked at the previous sister paper by Babkoff and Fostick (2013) and saw the figures in that paper are much clearer, and actually follow most of my suggestions.

Figure 1

As I understand it, the figure shows the number of ‘left leading’ responses, but those responses increase on the right side of the figure, which is counterintuitive. At a minimum I recommend indicating left and right on the x axis. Replotting the data more intuitively would be better.

Top panel

1) move the lower half of y axis values to the opposite side of the axis so the values are not obscured by the lines (or possibly move the entire y axis to the left of the figure)

2) order the colors of the lines systematically, so the relationship between tone duration and performance is easier to unpack

3) add the n per group to the figure, possibly under the tone duration…or at least to the caption

4) give an estimate of the error…one possibility would be to plot one point and a mean error bar for each duration in the upper left quadrant

5) add schematic diagrams illustrating the distinction between panel a and panel b

Bottom panel

1) use the same (revised) colors from the top panel for the points in this panel, for continuity, and to help the reader see the connection between the two panels

Figure 4

Same comments as for Figure 1.

Top panel

1) use the same (revised) colors for the different durations as in Fig. 1, for continuity

Figures 2,3 and 5,6

Same basic comments as for Figure 1

I recommend combining Figures 2 and 3 into one two-panel figure, and combining Figures 5 and 6 into another two-panel figure.

Grammar and word choice

Pg. 3, top: “reflects the individual’s ability”�”reflects an individual’s ability”

Pg., 3, middle: “and not by other cues”�” and not on other cues”

Pg. 3, bottom: “offset of the tone” “offset of the first tone”

Pg. 6, top: I recommend: “(hearing thresholds of 20 dB HL or less in frequencies 500, 1,000,

2,000, and 4,000 Hz)” “(hearing thresholds of 20 dB HL or less at 500, 1,000, 2,000, and 4,000 Hz)”.

Pg. 6, bottom: “2 order of presentation to each ear”…there is something odd about the grammar …possibly “2 presentation orders”

Pg. 6, bottom: [see18]�[see 18]

Pg. 7, top: “using the Danplex DA64 audiometer” using a Danplex DA64 audiometer.

Pg. 7, bottom, and throughout manuscript: Put a period after “Fig”

Pg. 7, bottom: “for the data plotted, as a function of SOA” “for the data, plotted as a function of SOA”

Pg. 7, bottom: “mean data was tested” “mean data were tested”

Pg. 8, middle: “When data from all 226 participants was analyzed” When the data from all 226 participants were analyzed”

Pg. 8, bottom: “The data from Experiment 1 shows that” “The data from Experiment 1 show that” (data is a plural word)

Pg. 8, bottom: “The extension of the tone duration from 10-40 msec” “The extension of the range of tone durations from 10-40 msec”

Pg. 9, middle: “to the TOJ threshold, we do not know” “to the TOJ threshold in aging adults, we do not know”

Pg. 10, bottom: I recommend: “(hearing thresholds of 35 dB HL or less in frequencies 500, 1,000, 2,000, and 4,000 Hz)” “(hearing thresholds of 35 dB HL or less at 500, 1,000, 2,000, and 4,000 Hz)”.

Pg. 12, bottom: “parameter in determining”�”parameter predicting”

Pg. 13, middle: “from 10-to-3 msec” “from 10 to 3 msec”

Pg. 13, middle: “than just serve to separate the onset of the first from the second tone”�” than simply serving to separate the onsets of the first and second tones.”

Pg. 14, bottom: “Theoretically, our finding shows that”�”Theoretically, our findings show that”

Pgs 22 and 25: The caption styles differ slightly.

6. PLOS authors have the option to publish the peer review history of their article (what does this mean?). If published, this will include your full peer review and any attached files.

Reviewer #1: No

Reviewer #2: No

---

## [Author Response · Author response to Decision Letter 0]

4 Jan 2022

Dear Prof. Nittrouer,

I would like to thank you for allowing a resubmission of a revised version of the manuscript “The role of tone duration in dichotic temporal order judgment II: Extending the boundaries of duration and age.” I would like to thank the reviewers for the useful comments that gave us the opportunity to make the manuscript clearer. Detailed here, as an accompaniment to the revised manuscript, is a summary of the changes made in response the reviewers’ suggestions. The page and line numbers referred to the “manuscript” version (clean version, with no track changes).

Reviewer #1:

1. The authors conclude that tone duration, then, just provides more information about SOA. It seems to fit the idea that stimulus onsets are very important for perception. A question of significance and motivation for the study arises: is this new information that contributes significantly to what we know? In asking that, it seems that the motivation for the use of shorter duration stimuli isn’t strong. I’m not sure why we expected that the results from 3-ms stimuli would be different from 10-ms stimuli (and from Fig 1 perhaps they are). If the stimuli are appropriately ramped, then spectral splatter should be minimized and one might expect this result. What were the temporal ramps applied to the stimuli? 

Response. As requested by the reviewer, we have now clarified in the Task and stimuli section that rise/fall times were 1 msec. Also, in an answer to the reviewer’s question as to why we expected shorter stimuli might produce different results than a longer one, we added our rationale to the Introduction section, namely that the duration of short sounds affects its perception. 

On page 5, line 14, we note:

“Our finding, however, is limited to the range of stimulus durations we tested, namely tone durations of 10 – 40 ms. It is unclear whether this finding would extend to other tone durations, since sound duration affects our auditory perception in several ways. First, sound duration affects the loudness of a sound via temporal summation, with sounds being perceived as softer or louder when duration is decreased or increased (respectively), up to 200 msec [33]. Second, sound duration affects our ability to perceive pitch, with lower frequencies requiring longer sound durations than higher frequencies. Third, sound duration also affects our ability to localize a sound source, with longer sounds being localized better by allowing the listeners to move their head towards the sound source [33,34].”

2. Also, based on Figure 1a. it appears that the 3 msec stimuli do separate themselves from the other data, with flatter functions.

Response. There was a difference between all stimulus duration, but only for short ISIs (ISIs 5 – 30 msec) not for longer ones (ISIs 60 – 240 msec). Following the reviewer’s comment, We added to both studies repeated-measures ANOVAs on accuracy data, with Stimulus Duration as a between-subjects variable and ISI as within-subjects variable. The results for Experiment 1 showed a main effect for Stimulus Duration, and a Stimulus Duration X ISI interaction. The source of this interaction was an effect for Stimulus Duration only for short ISIs (ISIs 5 – 30 msec) not for longer ones (ISIs 60 – 240 msec), as was also the case in Babkoff and Fostick (2013) study. 

In the Results section of Experiment 1 (page 10, line 16), we note:

“The accuracy data were transformed by probit (transformation for linearizing sigmoid distributions of proportions [50]. Psychometric functions of the probit-transformed data for the proportion of 'left leading' responses, as a function of ISI, are presented in Figure 2a, separately for each of the eight stimulus durations. A two-way repeated measures analysis of variance (ANOVA) was performed with the probit-transformed data as the dependent variable, ISI as a within-subjects variable, and Stimulus Duration as a between-subjects variable. The analysis revealed main effects of both ISI [F(7,1358) = 948.142, p < .001, ηp2 = .830] and Stimulus Duration [F(7,194) = 6.456, p < .001, ηp2 = .189], as well as an ISI × Stimulus Duration interaction [F(49,1358) = 1.670, p = .003, ηp2 = .057]. Post-hoc ANOVAs between Stimulus Duration for each ISI revealed significant effects of stimulus duration at the short ISIs [5 msec: F(7,194) = 4.288, p < .001; 10 msec: F(7,194) = 5.038, p < .001; 15msec: F(7,194) = 9.599, p < .001; 30 msec: F(7,194) = 5.573, p < .001], but not at the longer ISIs (60, 90, 120, and 240 msec; ps > .05).”

In the Results section of Experiment 2 (page 14, line 9) we note:

“The accuracy data were transformed by probit (transformation for linearizing sigmoid distributions of proportions [50]. The psychometric functions of the probit-transformed data for the proportion of 'left leading' responses, as a function of ISI, are presented in Figure 4a, for each of the four stimulus durations. A two-way repeated measures ANOVA was performed with the probit-transformed data as the dependent variable, ISI as a within-subjects variable, and Stimulus Duration as a between-subjects variable. The analysis revealed main effects for the ISI [F(7,651) = 244.105, p < .001, ηp2 = .724] and an ISI × Stimulus Duration interaction [F(28,651) = 3.291, p < .001, ηp2 = .124], but no main effect for the Stimulus Duration [F(4,93) = 1.165, p = .332, ηp2 = .048]. Post-hoc ANOVAs of Stimulus Duration for each ISI revealed significant effects of stimulus duration at some ISIs [5 msec: F(4,93) = 5.647, p < .001; 15msec: F(4,93) = 6.879, p < .001; 240 msec: F(4,93) = 4.714, p = .002], but not most (10, 30, 60, 90, and 120 msec; ps > .05).”

3. The reference for spectral splatter of short duration tones is quite out-dated (1967) and presumably instrumentation has changed dramatically since that time. Suied et al from JASA in 2014 have more recent data, and there are probably others as well.

Response. We agree with the reviewer and, as suggested, updated our citations on this topic to include Suied et al. (2014) and the following additional two newer references: 

Beatini JR, Proudfoot GA, Gall MD. Effects of presentation rate and onset time on auditory brainstem responses in Northern saw-whet owls (Aegolius acadicus). The Journal of the Acoustical Society of America. 2019 Apr 17;145(4):2062-71. 

Lee C, Guinan Jr JJ, Rutherford MA, Kaf WA, Kennedy KM, Buchman CA, Salt AN, Lichtenhan JT. Cochlear compound action potentials from high-level tone bursts originate from wide cochlear regions that are offset toward the most sensitive cochlear region. Journal of neurophysiology. 2019 Mar 1;121(3):1018-33.

4. It does seem interesting to address the question of the aging auditory system. Perhaps one might hypothesize that the only differences between younger and older listeners in TOJ would be found with short duration stimuli. This is introduced on page 10. However, then the older adults were tested using 10 to 40 ms stimuli, so it’s challenging to connect experiments 1 and 2. It does not appear that the age-related differences are duration specific, and that seems like an important point that could be emphasized more strongly. I think that question could be addressed more directly in the figures and in the analysis.

Response. As suggested by the reviewer, we now put more emphasize on the hypothesis of age-related differences in processing short stimuli, and on the conclusion that duration is not specific to age-related differences. Accordingly, we made the following revisions:

a) We added specific mention of the aging adults’ results in the abstract.

On page 2, line 16, we note:

“The results of both experiments confirmed the hypothesis, that the SOA required for performing dichotic TOJ was constant regardless of stimulus duration, for both age groups: about 66.5 msec for the young adults and 33 msec longer (100 msec) for the older adults.” 

b) We put the literature review on age-related differences in auditory temporal processing earlier in the Introduction section. 

On page 6, line 21, we note:

“Indeed, older adults have difficulty processing short and rapid stimuli. This difficulty is often reflected in the difficulty of older adults in perceiving speech, especially when the speaker talks fast or when speech is accompanied by background noise [3,17,21,24,39-41]. Studies of temporal processing among older adults, including the studies using speech stimuli, have reported deficiencies in their performance compared to young adults [5,7,14,21,42-49]. These studies demonstrated that older adults required longer sound durations [42,43,47], ISIs [5,7,14,21,45], and longer gaps within sounds [44,46,48,49], than young adults, in order to correctly perceive them. Such findings suggest that older adults might be sensitive both to tone duration and ISI. 

However, since no study has directly measured the mutual contribution of these two variables to the TOJ threshold in older adults, follow-up questions arise: Do older adults extract the same temporal information from the stimulus duration as from the ISI, so that each of these variables has the same effect on their dichotic TOJ threshold, as is the case for younger adults? Or do older adults extract different temporal information from the stimulus duration than from the ISI, so that each of these variables has different effects on their resulting dichotic TOJ threshold? If the pattern of TOJ performance by older adults is similar to that of younger adults, a “zero” line slope would be expected when thresholds are plotted as a function of SOA. This “zero” line slope is expected for both populations, although the thresholds for older adults are expected to be longer than that of younger adults due to age-related temporal deficits among older adults. However, if older adults have greater difficulty processing shorter duration than longer duration tones, the slope of the line relating the dichotic TOJ threshold to tone duration should be significantly greater than “zero”, indicating a greater contribution to dichotic TOJ threshold of tone duration than just the increase in SOA. To address these questions, in the current study we repeated the dichotic TOJ study using the same design as Babkoff & Fostick [1] but with participants whose ages ranged from 60-75 years.”

c) We have now elaborated the conclusions regarding older adults in the Summary and Conclusions section. 

On page 18, line 16, we note:

“Several theoretical and clinical conclusions arise from the experiments carried out in the present study. Our first conclusion is that when judging the order of two tones presented to the two ears, individuals extract the same temporal information from the stimuli as from the silent gap between them, whether they are young and have intact temporal processing abilities or are older and have less than intact temporal processing abilities.”. 

5. I have questions about the fit in Figure 2. I believe readers will need more information about that. Visual inspection shows a great deal of variability and thus it is difficult to understand the proportion of variance that is explained by the linear fit. Was heterogeneity tested? More details are need for me, and presumably other readers, to understand the data.

Response. We appreciate the opportunity to clarify this. The standard error of the data presented in this figure (Figure 2 in the original submitted version, Figure 3a in the current revised version) is 5.3 – 7 msec, and is also presented in the Results section of Experiment 1 (page 13, line 13). Heterogeneity testing was found to be non-significant (Levene Statistic, F(7,316) = 1.842, p = .079). In line with the reviewer’s comment, we added the results of the heterogeneity test to the Results section. 

On page 13, line 21, we note:

“ISI thresholds are presented in Figure 3a as a scattergram for all participants plotted against each tone duration. These data ranged correspondingly between 58.4 – 25.4 msec, with stable standard errors in the range of 5.3 – 7 msec. Heterogeneity testing (Levene Statistic) was not significant (F(7,316) = 1.842, p = .079).”

6. Fig 5. Is there a relationship between age and threshold among the older listeners? Age could be considered as a continuous variable.

Response. In response to the reviewer’s question, we checked for a correlation between age and threshold, but it was non-significant (r = 0.13, p = 0.21). 

7. In the Discussion section the authors state: as predicted the older adults’ thresholds were 33 ms longer than younger? Where was this predicted?

Response. As correctly pointed out by the reviewer, we predicted longer thresholds for older adults, but not in a certain number of msec. Correspondingly, we rephrased this sentence. 

On page 1, line 7, we note:

“As predicted, the thresholds for older adults were longer than for the younger adults; on average, the older adults’ TOJ threshold was longer by 33 msec across a 10 – 40 msec range of stimulus durations (i.e., an average SOA of 99.6 msec for the older adults vs. 66.5 msec for the younger adults).”

8. Data from younger and older listeners do appear from these data to be qualitatively similar, but it’s not clear enough yet why this is important. It is also unclear how experiment 1 fits into the overall picture. Additional motivation is needed, and details are needed about the stimulus characteristics (ramps) and the statistical analyses.

Response. As suggested by the reviewer here and in the previous comments, we put more emphasis on the study motivation and rationale, as well as elaborated on the conclusions and their potential contribution. We also added further detail such as the ramp information.

Reviewer #2:

1. Stimulus description: I think it is essential to include a description of how the threshold is determined (see other general comment), to include a figure illustrating how manipulating the tone duration allows the determination of whether the dichotic TOJ threshold is determined by the ISI or the SOA, and to include stimulus schematics in each figure to illustrate how the data are being analyzed (based on ISI or SOA). It took me quite a while to understand the stimuli. For example, here is a version of a note to myself as I was reading: Text [Pg. 3, bottom]: “We asked whether changes in the duration of the tones and inter-stimulus interval…affect dichotic temporal order judgment accuracy in the same or different ways” My note: “To me, it is confusing to introduce ISI here, because, I suspect, the TOJ threshold is determined by manipulations of the ISI…Oh!! Is the idea that the ISI is a silent period between the stimuli, but that in the manipulation of stimulus duration, the two tones are always contiguous??” Now I understand, or think I understand, that there are ISIs of various lengths between the two tones for all of the stimuli, that the stimulus duration was varied across conditions (with multiple ISIs for each stimulus duration), and that the question was whether the data are better explained as a whole by evaluating performance based on the ISI or on the stimulus onset asynchrony (SOA). This is a case where a picture really would be worth a thousand words.

Response. We agree with the reviewer that a picture really is worth a thousand words, so added the following illustration as Figure 1.

Figure 1. Schematic illustration of study design demonstrating the relationship of stimulus duration, inter-stimulus interval (ISI), and stimulus-onset asynchrony (SOA) (see top line). The manipulation of stimulus duration is presented as the duration of Stimulus A and Stimulus B, which varied across groups in the current study as a between-subjects variable. ISI is presented as the silent gap between the offset of Stimulus A and the onset of Stimulus B, which varies within each group as a within-subjects variable. Numbers of participants in each group for Experiments 1 and 2 are shown.

2. Threshold estimate: Why is the estimate of the time required between the onset of the two tones to determine their order—around 67 ms for younger adults—so much longer than the 15-20 ms value reported by Hirsh (1959)…a classic paper on temporal order judgment that is not cited in the current manuscript? Hirsh (and subsequently many others) used two ~500-ms stimuli whose onsets differed but whose offsets were coincident, but if temporal order is determined by stimulus onset asynchrony then it seems that the results of the present experiment and Hirsh’s data should align. Is the difference due to monotic vs. dichotic presentation?

Response. We very much appreciated this feedback and added references to the seminal work of Hirsh (1959) and Hirsh and Sherrick (1962) that were regretfully omitted. We also now reference later studies that found longer thresholds. On Page 4, line 17, we now note:

“Hirsh [28] and Hirsh and Sherrick [29] who measured the amount of time between the onsets of two stimuli (tones, clicks, lights, and their combinations) necessary to correctly report their order. This measure, called the TOJ threshold, reflects the minimum amount of time separating the onsets of the two stimuli at which an individual can correctly identify the order of stimulus presentation 75% of the time. Hirsh and Sherrick [29] originally reported the threshold for TOJ to be 17 msec, regardless of the type of stimulus and presentation modality used [29]. However, more recent studies have, generally, reported longer thresholds [1-3,5-7,10-13-18,30-32].”

3. Comment. Terminology: The terms temporal order judgment (TOJ) and dichotic TOJ are used interchangeably throughout the manuscript. It would be helpful to select just one term and then stick with it. Is the idea that the dichotic TOJ is just one more example of TOJ or that the dichotic aspect is an important factor? If the focus is on dichotic TOJ, the current claims could be tested with monaural TOJ tests, as well.

Response: The study was carried out on dichotic/spatial TOJ, as was the previous one. Following the reviewer’s comment, we added a paragraph in the Summary and Conclusions explaining why we chose this paradigm, and consequently the limitations of our findings and conclusions (see below). As suggested by the reviewer, we revised the manuscript to be more coherent when using TOJ / dichotic TOJ terminology. 

In the Summary and Conclusions, page 18, line 4, we note: 

“The present and the previous study [18] were both conducted on auditory dichotic TOJ (also referred to as spatial or binaural TOJ). This TOJ paradigm involves two identical sounds, presented asynchronously to the right and left ears. Other auditory TOJ paradigms involve two tones that differ in pitch or spectrum and are presented monaurally or diotically (to both ears at the same time). The advantage for measuring temporal processing using dichotic TOJ is that the stimuli are identical, and provides assurance that the temporal judgment is based on the temporal relationship of the two stimuli alone and not on other cues, such as pitch [19-20,27]. Furthermore, the perception of the stimulation of two ears by two asynchronous sounds, by definition, reflects mainly central auditory processing [1,18,28]. However, the conclusions drawn from the current and earlier studies are limited to this paradigm that was shown to mainly involve temporal cues [1,5,17,19,20]. The extent to which these conclusions can be generalized to other TOJ paradigms is yet to be tested.”

4. Writing: I found most of the manuscript to be quite difficult to read. The exception was the Summary and Conclusion section. The previous sister paper by Babkoff and Fostick (2013) is much clearer overall, indicating that the authors are capable of producing clear prose and therefore could greatly improve the clarity of the current manuscript.

Response. We appreciate the reviewer’s appreciation of our previous publication and took this constructive criticism very seriously. Accordingly, we thoroughly rewrote the manuscript as suggested in this comment and the following comments.

5. Introductions to individual experiments: I recommend combining all of the introductions into a single introduction. As it is now, some portions of the introductions to the individual experiments repeat information in the main introduction and other portions provide information that would be quite helpful to include in the main introduction.

Response. As suggested by the reviewer, the introduction sections of Experiment 1 and Experiment 2 were combined into a general introduction.

6. Results: I found the results section of the sister paper by Babkoff and Fostick (2013) to be much clearer and more informative than the current results sections. I recommend modeling the current results sections after the earlier paper while still incorporating new additions like the Bayes factor.

Response. As suggested, we have now modeled the reporting style of the results in the current paper after our previous publication (Babkoff and Fostick, 2013).

7. Stimulus duration: It would be quite helpful to spell out why the extension of the investigation to stimulus durations beyond 10-40 ms only focused on stimuli in the 3-8 ms range, rather than on a much wider range of stimulus durations. I think the reason is that the dichotic TOJ threshold is around 60 ms, so to compare ISI and SOA requires durations shorter than 60 ms. There is some mention of the possibility that the spectrum of the shortest stimuli would affect the outcome, but I did not find that argument to be compelling, especially without placing the 3-8 ms restriction in the larger context.

Response. To address the reviewer’s helpful feedback, we added further explanations regarding our focus on short durations. 

On Page 5, lines 14, we note:

“Our finding, however, is limited to the range of stimulus durations we tested, namely tone durations of 10 – 40 ms. It is unclear whether this finding would extend to other tone durations, since sound duration affects our auditory perception in several ways. First, sound duration affects the loudness of a sound via temporal summation, with sounds being perceived as softer or louder when duration is decreased or increased (respectively), up to 200 msec [33]. Second, sound duration affects our ability to perceive pitch, with lower frequencies requiring longer sound durations than higher frequencies. Third, sound duration also affects our ability to localize a sound source, with longer sounds being localized better by allowing the listeners to move their head towards the sound source [33,34]. 

The design of our study directed us toward testing tone durations shorter than those we used in our earlier study. The dichotic TOJ ISI threshold was found to be around 60 msec in several studies [1-3,9,11,16,18,21], therefore, manipulating tone duration, ISI and SOA necessarily places an upper limit on the tone durations one can test, i.e., they must be shorter than 60 msec. This means that in order to expand the range of tone durations necessary to test the generalization of our ISI-tone duration TOJ equivalence hypothesis, we focus on shorter durations than those we used in the previous study [1] (i.e., less than 10 msec). Such short durations can create transients (short-duration sounds with high amplitude that can accompany the beginning of short sounds) that spread energy across the frequency range [35-38], possibly resulting in different ISI-duration patterns than those observed with tone durations longer than 10 msec. Therefore, in the present study we aimed to test whether our finding applies to very short tone durations (i.e., 3, 6, and 8 msec) as well as durations in the 10-40 msec range, while using the same dichotic TOJ design as Babkoff & Fostick [1].”

8. TOJ threshold: The TOJ threshold is not defined.

Response. We have now added the definition of TOJ thresholds: in the Introduction, as presented by Hirsh (1959) and Hirsh and Serrick (1962), in the Task and Stimuli section of Materials and Methods, and in both of the Results sections.

On Page 4 line 17, note:

“TOJ has been studied extensively, beginning with the seminal work of Hirsh [28] and Hirsh and Sherrick [29] who measured the amount of time between the onsets of two stimuli (tones, clicks, lights, and their combinations) necessary to correctly report their order. This measure, called the TOJ threshold, reflects the minimum amount of time separating the onsets of the two stimuli at which an individual can correctly identify the order of stimulus presentation 75% of the time.”

In the Task and Stimuli section, page 9, line 14, we note:

“Dichotic TOJ thresholds were defined as the ISI necessary for 75 % accuracy, estimated using the best linear approximation of a psychometric function.”

In the Results section, page 11, line 20, we note:

“TOJ ISI thresholds, defined as the ISI necessary for 75% accuracy, were estimated using a linear function.”

In the Results section, page 1, line 5, we note:

“TOJ ISI thresholds (i.e., ISI required for 75% correct responses) for the older adult cohort…”

9. Comment. Terminology: The terms temporal order judgment (TOJ) and dichotic TOJ are used interchangeably throughout the manuscript. It would be helpful to select just one term and then stick with it. Is the idea that the dichotic TOJ is just one more example of TOJ or that the dichotic aspect is an important factor? If the focus is on dichotic TOJ, the current claims could be tested with monaural TOJ tests, as well.

Response. We appreciate the opportunity to clarify. Dichotic TOJ is the form of TOJ used in the present study, as in the previous one. To be more precise in this revised version of the manuscript, we now mention more explicitly that we use this paradigm, and, as suggested by the reviewer, we have corrected those places where it wasn’t mentioned specifically. We agree with the reviewer that our findings could be tested with monaural TOJ tests as well, therefore we also added to the Summary and Conclusions limitations regarding the use of this specific paradigm and the ability to generalize findings from this paradigm to other TOJ paradigms. 

On page 19, line 15 – page 20, line 3, we note:

“The present and the previous study [1] were both conducted utilizing auditory dichotic TOJ (also referred to as spatial or binaural TOJ). This TOJ paradigm involves uses two identical sounds presented asynchronously to the right and left ears; other auditory TOJ paradigms use two tones that differ in pitch or spectrum and are presented either monaurally or diotically (to both ears at the same time). The advantage of measuring temporal processing using dichotic TOJ is that the stimuli are identical, providing assurance that the temporal judgment is based on the temporal relationship of the two stimuli alone and not on other cues such as pitch [2,3,15]. Furthermore, perception of the stimulation of both ears by two asynchronous sounds, by definition, reflects mainly central auditory processing [1,16,51]. Consequently, the conclusions drawn from the current and earlier studies are limited to TOJ as tested by this paradigm, which has been shown to mainly involve temporal cues [2,3,9,11,16,]. The extent to which these conclusions can be generalized to other TOJ paradigms has yet to be tested.”

10. Specific Comments: Abstract. Pg. 2, top: “the major predictor of auditory TOJ threshold, and performance on spatial/dichotic TOJ tasks” Is there a difference between the TOJ threshold and performance on TOJ tasks? Is the intent that the predictor is for TOJ tasks in general, and the present results are for dichotic TOJ tasks in particular?

Response. The task tested in the current and previous study was spatial/dichotic TOJ. In previous studies, we showed different response patterns for different types of TOJ tasks, so it is important to state the specific TOJ that is tested here. However, as was correctly pointed out by the reviewer, the phrasing of the sentence was confusing, so we changed it accordingly. 

In the beginning of the abstract (page 2, line 1), we note:

“Temporal order judgment (TOJ) measures the ability to correctly perceive the order of consecutive stimuli presented rapidly. Our previous research suggested that the major predictor of auditory dichotic TOJ threshold, a paradigm that requires the identification of the order of two tones, each of which is presented to a different ear, is the time separating the onset of the first tone from the onset of the second tone (stimulus-onset-asynchrony, SOA).”

11. Pg. 2: To me, the abstract as a whole does not capture the major message of the manuscript—that dichotic TOJ thresholds appear to be determined by the SOA, rather than the ISI. I think the abstract would be much stronger if it were introduced using the argument at the beginning of the Summary and Conclusions about how the manuscript provides two tests of the idea that it is the SOA rather than ISI that determines the dichotic TOJ threshold.

Response. We accept the reviewer’s feedback and have now implemented the wording used in the beginning of the Summary and Conclusions, in order for the abstract to better capture the major message of the manuscript. 

In the abstract (page 2, line 8), we note:

“The current study aimed to evaluate the generalizability of the earlier finding by manipulating the experimental model in two different ways: a) extending the tone duration range to include shorter stimulus durations (3 – 8 msec; Experiment 1) and b) repeating the identical testing procedure on a different population with temporal processing deficits, i.e., older adults (Experiment 2).”

12. Introduction. Pg. 3, top: “Temporal order judgment (TOJ) reflects the individual’s ability to correctly perceive the order of consecutive stimuli presented rapidly.” This sentence is not clear to me. It seems to conflate the task (temporal order judgment) with performance on the task (correctly perceive…and presented rapidly).

Response. In line with this comment, the word “reflects” was changed to “measures”. 

The first sentence of the introduction (page 4, line 2) now states:

“Temporal order judgment (TOJ) measures the individual’s ability to correctly perceive the order of consecutive stimuli presented rapidly.”

13. Pg. 3, top: “TOJ thresholds” What is a TOJ threshold? I think it would help to introduce the ideas a bit more slowly. For example, I suspect that the TOJ threshold is determined by varying the ISI.

Response: We appreciate the feedback and have tried to address it in the manuscript accordingly. We start first with the definition of TOJ threshold in the second paragraph of the manuscript, before using this term later. We also repeat the definition of TOJ threshold later in the Methods and Results section.

On Page 4 line 17, note:

“TOJ has been studied extensively, beginning with the seminal work of Hirsh [28] and Hirsh and Sherrick [29] who measured the amount of time between the onsets of two stimuli (tones, clicks, lights, and their combinations) necessary to correctly report their order. This measure, called the TOJ threshold, reflects the minimum amount of time separating the onsets of the two stimuli at which an individual can correctly identify the order of stimulus presentation 75% of the time.”

In the Task and Stimuli section, page 9, line 14, we note:

“Dichotic TOJ thresholds were defined as the ISI necessary for 75 % accuracy, estimated using the best linear approximation of a psychometric function.”

In the Results section, page 11, line 20, we note:

“TOJ ISI thresholds, defined as the ISI necessary for 75% accuracy, were estimated using a linear function.”

In the Results section, page 1, line 5, we note:

“TOJ ISI thresholds (i.e., ISI required for 75% correct responses) for the older adult cohort…”

14. Pg. 3, top: “TOJ thresholds were found to be related to phonological skills [1-10] and to speech perception [2, 8,11-16].” What was the direction of the relationship? I assume that higher thresholds were associated with poorer phonological skills and poorer speech perception, but it would be helpful to state that directly.

Response. As suggested, the direction of the relationship between TOJ thresholds and phonological skills and speech perception was added. 

This sentence (page 4, line 13) now states: 

“The auditory TOJ paradigms have been used in studies of language skills over the last few decades, and have reported that better TOJ performance is related to better phonological skills [10-12,16-22] and better speech perception [8,10,21,23-27].”

15. Pg. 3, middle: “use stimuli that differ in spectrum or in duration [17], in frequency (pitch)” I do not understand the distinction between spectrum and frequency. The spectra differ for sounds of two different frequencies. I suspect that intent is that ‘spectrum’ means the spectrum of a complex sound, but that is not clear from the text.

Response. Spectrum indeed refers to pure tone vs. noise. Frequency refers to the sounds’ specific pitch. As suggested, this information was added to enhance the clarity. 

This sentence (page 4, line 10) now states:

“auditory TOJ paradigms use stimuli that differ either in: a) frequency (pitch)[1-8]; or b) spectrum (pure tone vs. noise) [9]; or c) duration [9]; or d) the ear of presentation, i.e., the ear that receives the first and the ear that receives the second stimulus (referred to as dichotic, spatial, or binaural TOJ)[1-2,5-7,9-15].”

16. Pg. 3, middle: “or in synchronicity, meaning which ear receives the first/second stimulus (2,5,9,17-19; 22-23,24-27]” I do not understand the term ‘synchronicity’ in this context. Would ‘ear of presentation’ work? Does synchronicity mean the same thing as dichotic TOJ?

Response. We appreciate the reviewer’s questions. The term ‘synchronicity’ indeed refers to the dichotic TOJ so that in dichotic TOJ the sounds are delivered asynchronously. In order to provide greater clarity, we replaced the term “synchronicity” with “ear of presentation”. This sentence (page 3, lines 13-16) now states:

“auditory TOJ paradigms use stimuli that differ either in: a) frequency (pitch)[1-8]; or b) spectrum (pure tone vs. noise) [9]; or c) duration [9]; or d) the ear of presentation, i.e., the ear that receives the first and the ear that receives the second stimulus (referred to as dichotic, spatial, or binaural TOJ)[1-2,5-7,9-15].”

17. Pg. 3, bottom: “of the tones” what tones?

Response. This refers to the tones that constitute the TOJ task. To remove ambiguity, we have added a clarifying phrase. 

On page 5, line 3, we mote:

“In a previous study [1], we and others reported the sensitivity of the dichotic TOJ paradigm to methodological and stimulus parameters, specifically to stimulus duration. We considered the possibility that the two manipulations, tone duration and ISI, might affect perception differently, since increasing tone duration increases the amount of sound—thus, the amount of stimulation—at the two ears, while increasing the ISI increases the silent interval between these stimulations—i.e., the lack of stimulation.”

18. Experiment 1: Young Participants, Stimulus Duration 3 to 40 ms. Materials and Method

Pg. 5, bottom: “226 participants” It would be helpful to include the number of males and females. Were there any sex differences in performance? Pg. 6, top: “The age of the participants ranged from 20 – 35 years.” It would be helpful to add the mean and standard deviation of the ages. Pg. 6, top: Were the participants compensated for their participation?

Response. In response to these questions, more detailed information on the participants was added. The Participants section now states (page 8, line 17):

“Participants were 226 undergraduate students (136 females, 90 males), aged 20 – 35 years (mean = 25.5, SD = 2.8) who volunteered to participate in the study. The current analyses include participant data presented in the earlier paper (n = 65) [1] together with the data from an additional 161 participants (current study).”

19. Pg. 6, bottom: “ranging between 5-240 msec” It would be helpful to list the ISI values. Pg. 6, bottom: Were the participants given trial-by-trial feedback? Pg. 6, bottom: “to ascertain whether they perceived the order of the tones and correctly reported the ear stimulated (right or left)” The distinction between perceiving the order of the tones and correctly reporting the ear stimulated is not clear to me. How can one be done without the other? Is the idea that participants could tell that the two tones were presented in opposite ears, sequentially, but could not indicate which ear was first? Pg. 6: What was the level of the tones?

Response. The information requested by these questions was added to the manuscript accordingly. 

The Task and stimuli section now states (page 9, line 6):

“We used the experimental design reported in Babkoff and Fostick [1]. In short, participants were presented with two 1 kHz pure tones at a level of 40 dB SL. The tones were presented asynchronously to the right and left ear, and participants were asked to report the order in which they heard them (either right-left or left-right). The tone duration for each participant was 3, 6, 8, 10, 15, 20, 30, or 40 msec, according to their assigned group (between subjects design). Rise/fall times were 1 msec. Eight different ISIs of 5, 10, 15, 30, 60, 90, 120, and 240 msec were randomly used. Each ISI value was repeated 16 times, producing 256 trials (8 ISIs × 2 presentation orders × 16 repetitions). After every 32 trials, participants received a short break. Dichotic TOJ thresholds were defined as the ISI necessary for 75 % accuracy, estimated using the best linear approximation of a psychometric function.

The experiment was preceded by a training session performed with tones of the same duration as in the experiment. This was designed to familiarize participants with the sounds and to ascertain whether they correctly reported the ear that was being presented with the sound (right or left) [see 1]. Participants received feedback for their responses during training sessions, but no feedback was presented during the experiment.”

20. Pg. 7 middle: “All participants were screened for hearing difficulties and their absolute threshold for 1 kHz was measured using the same computer and headphones that were used in the study.” Was the screening at 1 kHz separate from the screening for normal hearing mentioned in the ‘Participants’ section?

Response. Indeed, these were two separate procedures. To further clarify it in the manuscript we rephrased this sentence. The Procedure and stimuli section now states (page 10, line 8):

“Participants were screened for normal hearing prior to the experiment, after signed informed consent was obtained. In addition, their absolute threshold for 1 kHz was measured using the same computer and headphones that were used in the study.”

21. Pg. 7, bottom: “Mean ISI thresholds” How is the ISI threshold defined?

Response. Thresholds were defined as the ISI necessary for 75% accuracy, estimated using a linear function. This definition has now been added before describing the results of ISI thresholds. The second paragraph of the Results (page 11, line 20) now states:

“TOJ ISI thresholds, defined as the ISI necessary for 75% accuracy, were estimated using a linear function.”

22. Pg. 7, bottom: “Group mean data was tested against the predicted model and was found not to deviate significantly from this model (Probit analysis, Z = -3.13; p = .002).” What was the predicted model?

Response. The predicted model was of a similar reduction in threshold as the increase in tone duration. This clarification was added accordingly. 

Page 11 line 1, now states:

“Group mean data are also plotted (Figure 3a) and were tested against a model that predicted a reduction in threshold for the same magnitude of increase in tone duration. The data were found not to deviate significantly from this model (probit analysis, Z = -3.13; p = .002).”

23. Pg. 7, bottom: “Mean ISI thresholds for the data collected in the present study (3 – 8 ms) and the previous study (10 – 40 ms) ranged between 58.4 – 25.4 msec” The impression I get from this sentence (and elsewhere) is that all of the new data (n=161) are for 3-8 ms, and all of the previous data (n=65) are for 10-40 ms, but that does not fit with the n provided for each duration separately. Pg. 7, bottom: “ranged between 58.4—25.4 ms” In which direction?

Response. For the current study, additional participants were recruited for the 10 – 40 msec groups, as well as new participants for the 3 – 8 msec groups. As the reviewer correctly pointed out, the current wording gave an inaccurate impression, so we have now changed it, along with adding the corresponding direction. 

Page 11, line21, now state:

“ISI thresholds are presented in Figure 3a as a scattergram for all participants plotted against each tone duration. These data ranged correspondingly between 58.4 – 25.4 msec, with stable standard errors in the range of 5.3 – 7 msec.”

24. Pg. 7, bottom: Fig. 1B, the points above the ~-250 ms and +250 ms SOA are out of line with the rest of the data. I think this pattern deserves mention in the results section…presumably arises because of a flattening of performance, asymptotic performance at the longest ISI values. What is the outcome if these values are removed from the line fitting? My sense is that these values lead to underestimation of the ‘true’ slope. Is there a reason for using linear vs. log values?

Response. As suggested, we added a description of the data for these extreme SOA values, and repeated the analysis when they are omitted. Page 11, line 14, now state:

“Notwithstanding, the points below and above SOAs of -200 and +200 msec were out of line with the rest of the data. This might be due to an asymptotic performance at the longest ISI values. Repeating the analysis without these values included resulted in a predictive value of 98.9% (y = 0.0103x - 6E-18).”

25. pg. 8, top “TOJ thresholds” How is the TOJ threshold defined? Is it the same as the dichotic TOJ threshold mentioned later in the same paragraph? Is it the same as the ISI threshold mentioned in the preceding paragraph?

Response. As suggested, TOJ thresholds (for dichotic TOJ measured in the present study) were defined in the beginning of the paragraph. 

Page 11, line 20, now states:

“TOJ ISI thresholds, defined as the ISI necessary for 75% accuracy, were estimated using a linear function.”

26. Pg. 8, top: “The best linear fit to the means is drawn as a straight line (R2= 0.69,

p<.001) and the predicted line based on y = a - bx is drawn as a dotted line.” What is the predicted line?

Response. The predicted line is based on y = a – bx, predicting a similar reduction in threshold as the increase in tone duration. This was added at the beginning of the paragraph and also repeated in the sentence referred to above. 

On page 1, line 1, we now state:

“Group mean data are also plotted (Figure 3a) and were tested against a model that predicted a reduction in threshold for the same magnitude of increase in tone duration. The data were found not to deviate significantly from this model (probit analysis, Z = -3.13; p = .002). The best linear fit to the mean ISI thresholds (R2= 0.69, p<.001) is depicted in Figure 3a (see straight line) as is the predicted line (based on y = a – bx, predicting a similar reduction in ISI threshold as the increase in tone duration; see dotted line).”

27. Pg. 8, bottom: “for a wide range of stimulus durations (3 – 40 msec)” I do not consider 3-40 ms to be a wide range of stimulus durations. I recommend deleting “for a wide range...” to the end of the sentence, and just stating the outcome.”

Response. Changed as suggested. This sentence (page 13, line 3) now states:

“The data from Experiment 1 show that, for stimulus durations of 3 – 40 msec, young adults utilize the same cue for temporal processing from both the stimuli (tone duration) and from the silent gap between them (ISI).”

28. Pg. 9, top: “suggest that the time between the onset of two tones that is required for perceiving their order is constant (around 60 – 70 ms)” I think this estimate is quite interesting, and deserves to be emphasized in the paper, discussion, and possibly abstract. However, I think that it would be better to point it out first in the results section (I had to go back to the figure to work out how the value was determined).

Response. We adopted the reviewer’s suggestion and now mention this finding in the abstract, the Results, and the Discussion. We also repeated in the Discussion the exact numbers and the way they are calculated. 

On page 2, line 14, we note:

“The results of both experiments confirmed the hypothesis, that the SOA required for performing dichotic TOJ was constant regardless of stimulus duration, for both age groups: about 66.5 msec for the young adults and 33 msec longer (100 msec) for the older adults.”

Page 13, line 10, now states:

“Moreover, as the average dichotic TOJ threshold crosses the vertical axis at approximately 57 msec (as found in our previous study [1]) to 70 msec (as in the current study, Figure 3a and b), this suggests that the time between the onset of two tones that is required for perceiving their order is constant (around 60 – 70 msec).”

29. Experiment 2: Elderly Participants, Stimulus Duration 10 to 40 msec. Materials and Method. Pg. 9, middle: “elderly” (mentioned several time): I recommend using a different term, possibly “older”

Response. This suggestion was accepted and “older adults” is now used throughout the paper.

30. Pg. 9, middle: “Studies of temporal processing among the elderly, including those researching it in the context of speech, have reported deficiencies in performance compared to young adults [8,19-20,22,24,26,38-45]. These studies have demonstrated that aging adults require longer sound durations, ISIs, and gaps, than do young adults, in order to correctly perceive them.” It would be helpful to indicate which of the references in the first sentence are associated with each of the measures listed in the second sentence.

Response. As suggested, we specified the appropriate studies for each measure. This sentence (page 7, line 1) now states:

“Indeed, older adults have difficulty processing short and rapid stimuli. This difficulty is often reflected in the difficulty of older adults in perceiving speech, especially when the speaker talks fast or when speech is accompanied by background noise [3,17,21,24,39-41]. Studies of temporal processing among older adults, including the studies using speech stimuli, have reported deficiencies in their performance compared to young adults [5,7,14,21,42-49]. These studies demonstrated that older adults required longer sound durations [42,43,47], ISIs [5,7,14,21,45], and longer gaps within sounds [44,46,48,49], than young adults, in order to correctly perceive them.”

31. Pg. 9, middle: “Such findings support the notion of aging adults’ sensitivity both to tone duration and ISI” I do not understand this phrase. Is the intent that such findings indicate that aging adults are sensitive to both…??

Response. This indeed is what we were suggesting, according to the findings of previous studies. In light of the reviewer’s comment, however, we rephrased this sentence to be clearer. This sentence (page 7, line 5) now states:

“Such findings suggest that older adults might be sensitive both to tone duration and ISI.”

32. Pg. 10, top: “If so, we expected the pattern of older adults’ performance to be similar to those of young adults, indicating that although the thresholds will be longer, on average, there will be a “zero” line slope when thresholds are plotted as a function of SOA.”�”If so, the pattern of older adults’ performance should be similar to that of young adults, such that there should be a “zero” line slope when thresholds are plotted as a function of SOA.” I recommend stating in a separate sentence why the thresholds are expected to be longer.

Pg. 10, middle: “than longer ones, we would expect the line’s slope”�”than longer ones, the line’s slope (relating dichotic TOJ threshold to tone duration) should be significantly greater than “zero”

Response. These suggestions were fully accepted. This paragraph, (page 7, line 13) now states:

“If the pattern of TOJ performance by older adults is similar to that of younger adults, a “zero” line slope would be expected when thresholds are plotted as a function of SOA. This “zero” line slope is expected for both populations, although the thresholds for older adults are expected to be longer than that of younger adults due to age-related temporal deficits among older adults. However, if older adults have greater difficulty processing shorter duration than longer duration tones, the slope of the line relating the dichotic TOJ threshold to tone duration should be significantly greater than “zero”, indicating a greater contribution to dichotic TOJ threshold of tone duration than just the increase in SOA.”

33. Pg. 10 bottom to pg. 11 top: I recommend taking out the subheads under Materials and Methods and just stating that Experiment 2 was the same as Experiment 1, except…(fill in the blanks).

Response. This suggestion was fully accepted. The Materials and Method section now includes the following (page 15, lines 3-9):

“Experiment 2 was conducted using the same methodology as Experiment 1, with the exception of: a) older participants and b) a duration range of 10-40 msec. A group of 98 participants (59 females, 39 males), aged 60 – 75 years (mean = 66.4, SD = 6.1), volunteered to participate in the study. Participants were divided into five groups, each of which was tested with only one tone duration, as follows: 10 msec (n = 16), 15 msec (n = 27), 20 msec (n = 17), 30 msec (n = 19), and 40 msec (n = 19). After providing signed informed consent, the participants were screened for age-normal hearing (hearing thresholds of 35 dB HL or less at 500, 1,000, 2,000, and 4,000 Hz). This was an inclusion criterion while hearing deficit was an exclusion criterion.”

34. Pg. 11 top: What does ‘resembled’ mean in this context? Were there a number of differences in the procedure between Experiment 1 and Experiment 2, but a vague similarity between the two experiments? I suspect not.

Response. The reviewer is correct, and this sentence has been omitted.

35. Pg. 12, middle: “the thresholds for aging adults were, on average, 33 ms longer than for the younger adults (an average SOA threshold of 99.6 ms for aging adults vs. 66.5 ms for the younger).” See comment above (pg. 9, top) about the estimate of the threshold. I recommend including that information first in the results section.

Response. As indicated in our previous response, we adopted the reviewer’s suggestion and now mention this finding in the abstract, the Results section, and the Discussion. 

The abstract now states (page 2, line 16):

“The results of both experiments confirmed the hypothesis, that the SOA required for performing dichotic TOJ was constant regardless of stimulus duration, for both age groups: about 66.5 msec for the young adults and 33 msec longer (100 msec) for the older adults.”

The Results now state (page 1, line 21):

“The point at which the average dichotic TOJ threshold (SOA) crosses the vertical axis in Figure 5b is 100 msec (probit analysis, Z = -2.74, p = .02), indicating that the TOJ threshold for the older adults is, on average, 33 msec longer than the young adults in Experiment 1 (99.6 msec vs. 66.5 msec, respectively).”

The Discussion now states (page 16, line 7):

“As predicted, the thresholds for older adults were longer than for the younger adults; on average, the older adults’ TOJ threshold was longer by 33 msec across a 10 – 40 msec range of stimulus durations (i.e., an average SOA of 99.6 msec for the older adults vs. 66.5 msec for the younger adults).”

36. Pg. 13, middle: “The present study aimed to evaluate the temporal mechanism of TOJ by testing the generalizability of the conclusion that dichotic TOJ is determined by stimulus onset asynchrony (SOA), the time separating the onset of the first tone to the onset of the second one. We tested this hypothesis by two different manipulations: 1) by extending the range of tone durations to include 3-10 msec on a population of young adults; 2) by performing identical testing on a population of older adults, about whom there is existing evidence of a general deficit in auditory temporal order judgment.” I think this set-up is much clearer than the one in the current introduction.

Response. We appreciate the reviewer’s positive feedback about this description. As suggested, we now utilize this wording in the Introduction. 

The Introduction now states (page 8, line 1):

“The aim of the present study was, therefore, to test our previous conclusion that dichotic TOJ is determined by stimulus onset asynchrony (SOA)—the time separating the onset of the first tone from the onset of the second one [1]—a) by using stimulus parameters that test its boundaries, and b) to determine if it can be generalized. We operationalized this aim by implementing two different manipulations to our previous research model: a) extending the range of tone durations to include also very short tone durations (3 – 8 msec) thus testing TOJ thresholds with tone durations ranging from 3 – 40 msec (Experiment 1), and b) using the same experimental methodology as in the previous study [1] to test a population of older adults (Experiment 2).”

37. Figure 1. As I understand it, the figure shows the number of ‘left leading’ responses, but those responses increase on the right side of the figure, which is counterintuitive. At a minimum I recommend indicating left and right on the x axis. Replotting the data more intuitively would be better.

Response. While we agree with the reviewer’s perspective that the figure design is somewhat counterintuitive, we would like to preserve this design in order to maintain consistency with figures in our previous paper (2013). However, to address the reviewer’s recommendation, we have now added indications of ‘Right leading’ and ‘Left leading’ on the x-axis, as we did in our previous paper, to enhance clarity.

38. Figure 1 top panel

1) move the lower half of y axis values to the opposite side of the axis so the values are not obscured by the lines (or possibly move the entire y axis to the left of the figure)

2) order the colors of the lines systematically, so the relationship between tone duration and performance is easier to unpack

3) add the n per group to the figure, possibly under the tone duration…or at least to the caption

4) give an estimate of the error…one possibility would be to plot one point and a mean error bar for each duration in the upper left quadrant

5) add schematic diagrams illustrating the distinction between panel a and panel b

Bottom panel

1) use the same (revised) colors from the top panel for the points in this panel, for continuity, and to help the reader see the connection between the two panels

Figure 4

Same comments as for Figure 1.

Top panel

1) use the same (revised) colors for the different durations as in Fig. 1, for continuity

Figures 2,3 and 5,6

Same basic comments as for Figure 1

I recommend combining Figures 2 and 3 into one two-panel figure, and combining Figures 5 and 6 into another two-panel figure.

Response. All suggestions were accepted fully. The revised figures are as follows (starting from Figure 2, as a new Figure 1 was added to illustrate the study design):

a. Data by duration and ISI

b. Data by SOA

Figure 2. Psychometric function of probit-transformed data for the proportion of 'left leading' responses of young adult participants across eight different stimulus durations for: (a) each stimulus duration by ISI; and (b) all data by SOA. Schematic diagrams illustrating the distinction between ISI and SOA appear in each panel.

 

a. ISI thresholds

b. SOA thresholds

Figure 3. TOJ thresholds of young adult participants across eight different stimulus durations for: (a) ISI thresholds (silent gap between the offset of the stimulus at the leading ear and the onset of the stimulus at the lagging ear); and (b) SOA thresholds (duration of the stimulus at the leading ear added to the ISI threshold).a. Data by duration and ISI

a. Data by duration and ISI

b. Data by SOA

Figure 4. Psychometric function of probit-transformed data for the proportion of 'left leading' responses of older participants across five different stimulus durations for: (a) each stimulus duration by ISI; and (b) all data by SOA. Schematic diagrams illustrating the distinction between ISI and SOA appear in each panel.

a. ISI thresholds

b. SOA thresholds

Figure 5. TOJ thresholds of older participants plotted as a function of stimulus duration for (a) ISI thresholds (silent gap between the offset of the stimulus at the leading ear and the onset of the stimulus at the lagging ear); and (b) SOA thresholds (duration of the stimulus at the leading ear added to the ISI threshold).

39. Comment. Grammar and word choice:

Pg. 3, top: “reflects the individual’s ability”�”reflects an individual’s ability”

Pg., 3, middle: “and not by other cues”�” and not on other cues”

Pg. 3, bottom: “offset of the tone” “offset of the first tone”

Pg. 6, top: I recommend: “(hearing thresholds of 20 dB HL or less in frequencies 500, 1,000,

2,000, and 4,000 Hz)” “(hearing thresholds of 20 dB HL or less at 500, 1,000, 2,000, and 4,000 Hz)”.

Pg. 6, bottom: “2 order of presentation to each ear”…there is something odd about the grammar …possibly “2 presentation orders”

Pg. 6, bottom: [see18]�[see 18]

Pg. 7, top: “using the Danplex DA64 audiometer” using a Danplex DA64 audiometer.

Pg. 7, bottom, and throughout manuscript: Put a period after “Fig”

Pg. 7, bottom: “for the data plotted, as a function of SOA” “for the data, plotted as a function of SOA”

Pg. 7, bottom: “mean data was tested” “mean data were tested”

Pg. 8, middle: “When data from all 226 participants was analyzed” When the data from all 226 participants were analyzed”

Pg. 8, bottom: “The data from Experiment 1 shows that” “The data from Experiment 1 show that” (data is a plural word)

Pg. 8, bottom: “The extension of the tone duration from 10-40 msec” “The extension of the range of tone durations from 10-40 msec”

Pg. 9, middle: “to the TOJ threshold, we do not know” “to the TOJ threshold in aging adults, we do not know”

Pg. 10, bottom: I recommend: “(hearing thresholds of 35 dB HL or less in frequencies 500, 1,000, 2,000, and 4,000 Hz)” “(hearing thresholds of 35 dB HL or less at 500, 1,000, 2,000, and 4,000 Hz)”.

Pg. 12, bottom: “parameter in determining”�”parameter predicting”

Pg. 13, middle: “from 10-to-3 msec” “from 10 to 3 msec”

Pg. 13, middle: “than just serve to separate the onset of the first from the second tone”�” than simply serving to separate the onsets of the first and second tones.”

Pg. 14, bottom: “Theoretically, our finding shows that”�”Theoretically, our findings show that”

Pgs 22 and 25: The caption styles differ slightly.

Response. Thank you for all of these corrections and suggestions. All comments were accepted, and wording was reviewed and updated throughout the manuscript as suggested by the reviewer. In addition, the shortening “Fig” was replaced with the full form of the word (“Figure”), and the caption of Figure 1 (page 22) was corrected to resemble that of Figure 4 (page 25), accordingly.

We hope you will find this revised version of the manuscript appropriate for publication in your journal.

Best regards,

Leah Fostick and Harvey Babkoff

---

## [Decision Letter · Decision Letter 1]

18 Feb 2022

The role of tone duration in dichotic temporal order judgment II: Extending the boundaries of duration and age

PONE-D-21-18616R1

Dear Dr. Fostick,

Thank you for submitting your work to PLOS ONE. I now have a review from one of the original reviewers. Although I was not able to obtain a review from the second reviewer, I am prepared to accept your manuscript based on this one review and my own assessment.

We’re pleased to inform you that your manuscript has been judged scientifically suitable for publication and will be formally accepted for publication once it meets all outstanding technical requirements. Thank you for your careful attention to the comments of the original reviews.

Kind regards,

Susan Nittrouer, Ph.D.

Academic Editor

PLOS ONE

Reviewers' comments:

Reviewer's Responses to Questions

**Comments to the Author**

1. If the authors have adequately addressed your comments raised in a previous round of review and you feel that this manuscript is now acceptable for publication, you may indicate that here to bypass the “Comments to the Author” section, enter your conflict of interest statement in the “Confidential to Editor” section, and submit your "Accept" recommendation.

Reviewer #1: All comments have been addressed

2. Is the manuscript technically sound, and do the data support the conclusions?

Reviewer #1: Partly

3. Has the statistical analysis been performed appropriately and rigorously? 

Reviewer #1: Yes

4. Have the authors made all data underlying the findings in their manuscript fully available?

Reviewer #1: (No Response)

5. Is the manuscript presented in an intelligible fashion and written in standard English?

Reviewer #1: Yes

6. Review Comments to the Author

Reviewer #1: Thanks to the author for their attention to previous reviews.

A minor note: I'm more familiar with using the abbreviation 'ms' for millisecond. I'll leave that to the editorial staff.

7. PLOS authors have the option to publish the peer review history of their article (what does this mean?). If published, this will include your full peer review and any attached files.

Reviewer #1: No

---

## [Editor Report · Acceptance letter]

22 Mar 2022

PONE-D-21-18616R1 

The role of tone duration in dichotic temporal order judgment II: Extending the boundaries of duration and age 

Dear Dr. Fostick:

I'm pleased to inform you that your manuscript has been deemed suitable for publication in PLOS ONE. Congratulations! Your manuscript is now with our production department. 

Kind regards, 

on behalf of

Dr. Susan Nittrouer 

Academic Editor

PLOS ONE